∂ | Open Peer Review | Microbial Ecology | Research Article

# Tracking putative *Microcystis* viruses and virus-host associations across distinct phases of a *Microcystis*-dominated bloom

A. J. Wing,[1] Bridget Hegarty,[1] G. Eric Bastien,[1] Vincent J. Denef,[1] Jacob Evans,[1] Gregory J. Dick,[1,2] Melissa B. Duhaime[1]

**ABSTRACT** Viruses significantly impact microbial community composition and function. Yet their role in the fate of freshwater cyanobacterial harmful algal blooms (cHABs), an increasing threat to freshwater systems, remains poorly understood. Here, we address this with a metagenomic analysis of viruses of bloom-forming *Microcystis aeruginosa* through a seasonal cHAB in the western basin of Lake Erie. We identified globally distributed *Microcystis* viruses in Lake Erie based on sequence homology to well-studied isolates. A machine-learning model was then used to predict associations between uncharacterized viral populations and the *Microcystis* and non-*Microcystis* hosts of the cHAB. Size fractionation of water samples allowed us to identify significant fraction-specific trends in *Microcystis* viral diversity that corresponded with *Microcystis* genetic diversity. Viral diversity was highest in the non-colony-associated fraction and lowest in the colony-associated fraction, suggesting that colony formation may lead to bottlenecks in viral diversity in cHABs. Significant turnover of predicted *Microcystis* virus populations was observed through time, but not between stations miles apart. The virus-host networks revealed extensive interconnectivity and the potential for virus-mediated cross-species genetic exchange. The networks predicted that Lake Erie *Microcystis* viruses infect hosts spanning phyla, in agreement with lab studies in other systems but challenging previous notions of "narrow" host-virus associations in this genus. Abundant *Microcystis* virus genes revealed a potential role in key metabolic pathways and host adaptation. These findings advance our understanding of *Microcystis* viruses and their potential influence on host metabolism, species interactions, and coevolution in *Microcystis*-dominated cHABs.

**IMPORTANCE** Understanding associations between viruses, their hosts, and environmental factors is key for identifying the mechanisms behind the rise and fall of cyanobacterial harmful algal blooms. This study explores the diversity and host ranges of viruses predicted to infect *Microcystis*, reporting how these properties vary over time, across sample stations in western Lake Erie, and among different filter size fractions. We found that *Microcystis* virus diversity is highest in non-colony-associated fractions and the lowest in colony-associated fractions, suggesting a link between *Microcystis* colony formation and reduced viral diversity. We identify abundant genes belonging to predicted *Microcystis* viruses and their potential roles in key metabolic pathways and adaptation to environmental changes. These findings enhance our understanding of the interplay among viruses, *Microcystis*, and co-occurring bacteria in cHABs, offering insights into the mechanisms driving bloom dynamics, species interactions, and coevolutionary processes.

**KEYWORDS** cyanophages, bacteriophages, microbial ecology, metagenomics, environmental microbiology, virology

Address correspondence to Melissa B. Duhaime, duhaimem@umich.edu.

The authors declare no conflict of interest.

See the funding table on p. 19.

*M*icrocystis aeruginosa is a colony-forming cyanobacterium that can form toxic cyanobacterial harmful algal blooms (cHABs) in freshwater and estuarine systems worldwide (1, 2). The toxins produced by *M. aeruginosa* (3), particularly microcystins, are toxic to humans and diminish drinking water quality and aquatic ecosystem health (4–6). The frequency and intensity of *M. aeruginosa* blooms are increasing (7, 8), largely due to climate change and eutrophication (5, 8–11). Given the ecologically-driven temporal succession of *Microcystis* strains (12–18) and *Microcystis* inter-species interactions (19–21), it is important to study these blooms as whole communities of co-existing, potentially interacting populations—such as offered through metagenomics. Further, while the role of abiotic controls (e.g., temperature, nutrients) on *Microcystis* bloom progression has been extensively studied, the role of biotic factors, such as predator-prey relationships, is less understood owing in part to the challenges of studying ecological interactions at the micron scale in complex community settings.

As microbial predators, viruses exert top-down controls on microbial communities by infecting and lysing microbial hosts (22–25). They also reprogram host metabolisms (26–31), facilitate gene transfer (32, 33), and can impact host fitness and ecosystem-level nutrient cycling through virus-encoded auxiliary metabolic genes (AMGs) (34–36). Previous research on isolated *Microcystis* viruses, such as the well-studied Ma-LMM01 (Japan) (37) and Ma-MVDC (China) (38), has provided foundational insights into virus-*Microcystis* interactions. There are eight additional, less well-studied *Microcystis* viruses isolated, sequenced, and/or described (39–46). However, our understanding of the spatiotemporal variation of *Microcystis* viruses is limited to only a few field studies in the United States/Canada, Japan, and China (2, 47–51 ). Further, these have primarily relied on (i) marker gene analysis that is insufficient for tracking associations between virus-host populations that lack these genes or (ii) CRISPR spacer matching between *Microcystis* and putative viruses, which excludes the nearly 90% of bacteria that lack a CRISPR-Cas system (52). While these studies have shed important light on the role viruses may play in *Microcystis* cHAB dynamics, an approach is needed that can consider ecological associations within the total community, as well as population variations at the whole-genome level, to build upon these foundations.

Evidence suggests that *Microcystis* in Lake Erie's annual cHABs is susceptible to viral infection and lysis (6, 48, 53). Release of microcystins from *Microcystis* cells due to viral lysis is thought to have contributed to the Toledo drinking water crisis in 2014 (6, 48). In 2019, a suspected cHAB lysis event in Lake Erie, that corresponded with the presence of putative *Microcystis* viruses, MVGF_J_19 and MVGF_J_348, may have contributed to the release of dissolved microcystins (48). Quantitative PCR and metagenomic analyses suggested that viral activity targeted specific *Microcystis* spp. within the 2019 Lake Erie bloom (48). These associations suggest that multiple lysis events may have occurred throughout the 2019 bloom (48, 54). Such findings suggest a role for viruses in Lake Erie's cHAB progression, yet the overall trends in viral community diversity, as well as the relationship between viral community features and the genetic composition of *Microcystis* populations, are not well characterized.

In this study, we used metagenomic analyses of size-fractionated viral and cellular communities in the western basin of Lake Erie to address the following. (i) How does the diversity and distribution of viruses that infect *Microcystis* change through time, across various sample locations, and among different filter size fractions—especially given the distinct genomic and physiological characteristics that may underlie free-living versus colonial *Microcystis* populations—during a cHAB? (ii) What potential impacts are *Microcystis* virus infections likely to have on host metabolism and gene flow within the *Microcystis* genus and other possible hosts?

## RESULTS

### Tracking *Microcystis* hosts in the 2014 Lake Erie seasonal cHAB

#### *Microcystis populations showed the greatest abundance and lowest overall evenness in colony fraction*

Water was sampled at three sites from the western region of Lake Erie (Fig. 1A), with collections bi-weekly in June and weekly from July to October in 2014. Concentrations of particulate phycocyanin (used as a proxy for cyanobacteria) and microcystin (indicative of bloom toxicity) revealed a toxic *Microcystis* bloom at all three sampling locations (WLE12, WLE2, and WLE4) in early August (Fig. 1A and B) (55). The cHAB in late September occurred primarily at the nearshore stations with lower microcystin concentrations than 4 August (Fig. 1B).

Metagenomic sequence data were generated from each station at the two bloom peaks and across five filter size fractions to separately enrich for *Microcystis* in free-living and colony forms, as well as their viruses (Fig. 1C). From these data, 50 bacterial metagenome-assembled genomes (MAGs) were reconstructed (Table S1). Twenty-six of the bacterial MAGs were taxonomically classified as *Microcystis* (Fig. 1D; Table S1). Seventeen of the *Microcystis* MAGs were from samples collected on 4 August and 29 September, the two bloom peaks (Fig. 1D; Table S1; Fig. S1). The distribution of relative abundances of *Microcystis* ranged from 0% to 29%, based on the proportion of the total reads in each sample that competitively mapped to the *Microcystis* MAGs in the metagenomes of the cellular fractions (0.22–100 µm; Fig. 1E; Fig. S1). At the bloom peaks, the highest relative abundances were observed in the 100 µm fraction (Fig. 1E). This colony-enriched fraction is also where the greatest overall variability of *Microcystis* MAGs was observed, which is reflected in the lowest overall evenness (Fig. 1F; Fig. S1; Table S2).

#### *Genetic variation of Microcystis populations corresponded with size fraction*

At the bloom peaks, distinct genetic differentiation of *Microcystis* populations was observed according to size fractions, based on both whole-genome nucleotide identity (Fig. 1D) and gene clustering analysis (Fig. 1G; Fig. S2). *Microcystis* MAGs from 100 µm filters clustered together based on ANI, with the exception of two *Microcystis* MAGs from late season 27 October (Fig. 1D). The *Microcystis* MAGs from the 100 µm fraction were taxonomically classified as four different species or strain types (Fig. 1D). *Microcystis* MAGs from the smaller size fractions were dominated by *Microcystis aeruginosa* C (GTDB annotation, which shares 95% ANI with *M. aeruginosa* NIES-2549 isolated from Lake Kasumigaura, Japan; Table S1) (56), with other variants occasionally observed (Fig. 1D). When the presence/absence of *Microcystis* MAG pangenome accessory gene clusters (defined as gene families present in at least 90% of the MAGs) (57) was considered, 33.3% of the variation could be explained by fraction, with the 100 µm MAGs forming a distinct grouping along principal component 1 (Fig. 1G).

In terms of the distribution of the pangenome gene clusters, a large proportion, 6,738 (78.3%), were shared in at least two fractions, and 3,529 (41%) were shared across all fractions. Some gene clusters were found in only one fraction: 1,572 (18.3% of total clusters) in the 100 µm fraction only, 12 (0.14%) in 53 µm, 133 (1.5%) in 3 µm, and 145 (1.7%) in >0.22 µm. Most of these fraction-specific genes are not functionally annotated; however, the two genes with functional annotations present in all 100 µm fractions were *lprI* (lysozyme inhibition) and *menH* (menaquinone biosynthesis; Table S3). The only gene with functional annotation in the not-colony-associated fraction was a restriction enzyme, *alwI* (Table S3).

### Identification of *Microcystis* viruses in the 2014 Lake Erie bloom

#### *Globally distributed Microcystis viruses identified through sequence homology*

From the total pool of contigs, 27,086 viral contigs >3 kb in length were identified, which were then binned and clustered into 15,461 non-redundant viral operational taxonomic

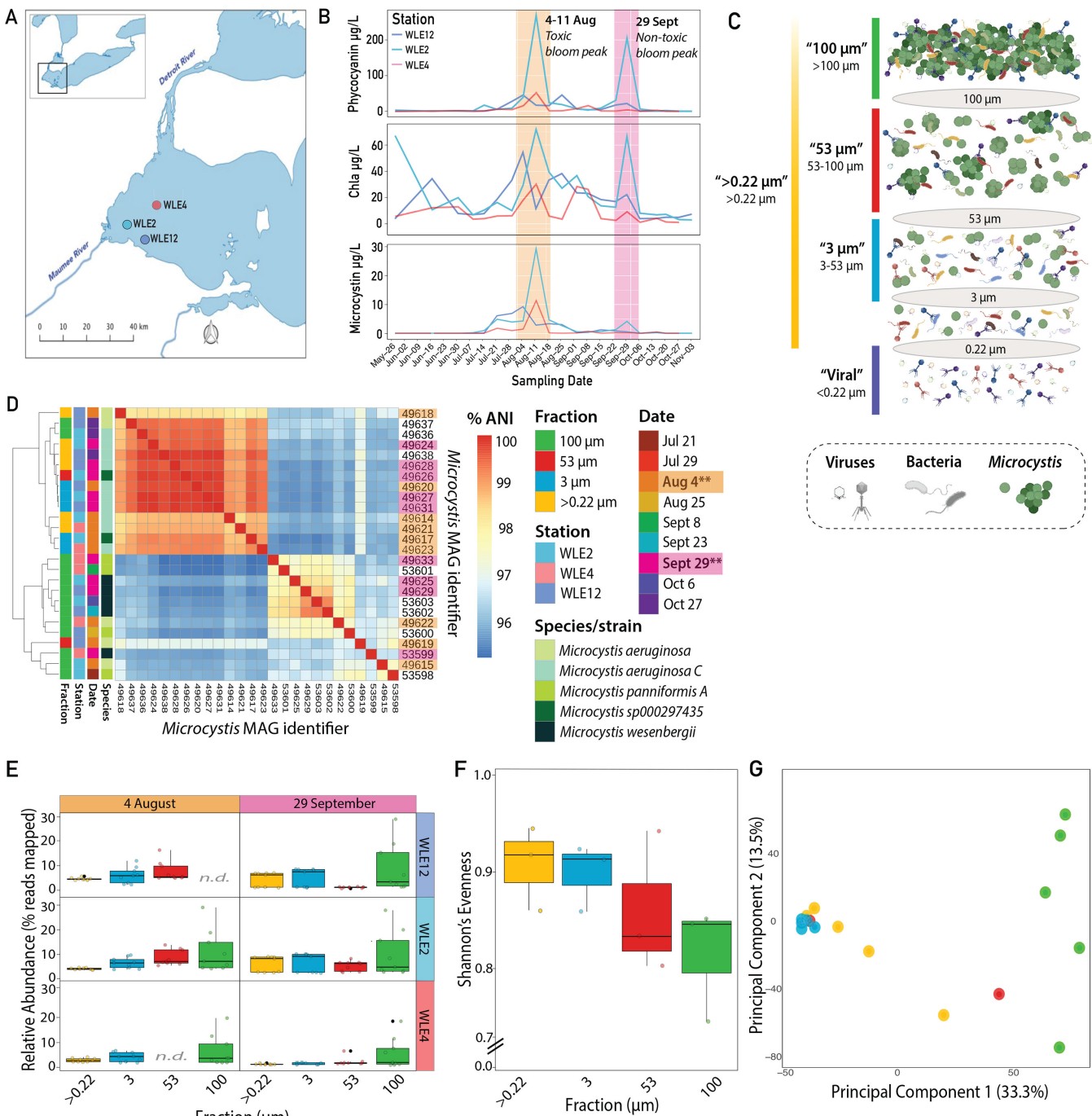

**FIG 1** Sampling overview and *Microcystis* MAG diversity and dynamics. (A) Map of sampling sites in the western basin of Lake Erie. (B) Phycocyanin, chlorophyll a, and microcystin measurements for the 2014 bloom by station. (C) Schema of sample collection using filter size fractionation. Viruses are depicted by multicolored virions, non-*Microcystis* bacteria by multicolored flagellated rods, and *Microcystis* by green cocci (as singlets, doublets, and colonies). (D) Heatmap of average nucleotide identity (% ANI) among 26 *Microcystis* MAGs from July to October across three stations and four sample fractions. Bottom row and right column list the MAG identifiers. Sample fraction, station, date, and species/strain-level taxonomic classification of MAGs are identified by color on left. A dendrogram depicts the clustering of MAGs based on %ANI. Asterisks near dates indicate bloom peaks. (E) Box and whisker plots of the distribution of relative abundances of 17 *Microcystis* MAGs from bloom peaks (9 MAGs on 4 August; 8 MAGs on 29 September). No data exist for 4 August 100 μm at WLE12 and 53 μm at WLE4. Points represent *Microcystis* MAGs. In all, boxes represent the middle 50% and whiskers represent the upper and lower quartile values in each fraction. (F) Box and whisker plots show Shannon's evenness measures for *Microcystis* MAGs across fractions at bloom peaks (4 August and 29 September). (G) PCA of bloom peak *Microcystis* pangenome accessory gene clusters by *Microcystis* MAG. Point color indicates which size fraction a *Microcystis* MAG belongs to and is consistent with fraction coloring in (C–F).

units (vOTUs), approximating viral species based on currently established thresholds of 95% ANI across 85% of the contig length (58). None of these viruses were integrated as prophages in the *Microcystis* MAGs. We identified four putative *Microcystis* vOTUs (vOTU_4, vOTU_1398, vOTU_4148, and vOTU_6227) with a high degree of similarity to four *Microcystis* virus isolates (Fig. 2; Table S4). The representative virus of vOTU_4, renamed here as Ma-LEF01 (*M. aeruginosa* Lake Erie Fukuivirus-01), had high similarity (>99% shared average nucleotide identity, ANI, across 95% of its genome) to a viral contig, MVGF-J19, that was previously assembled from a 2019 Lake Erie cHAB metagenome (Fig. S3) (48). MVGF-J19 and Ma-LEF01 were also highly similar to *Microcystis* isolates MaMV-DC and Ma-LMM01 (Fig. S3). All 7 of the 243 predicted genes that were unique to Ma-LEF01 were of unknown function (Table S5). While Ma-LEF01 carries genes that have been previously associated with both lytic (e.g., viral tail sheath) and lysogenic (e.g., putative phage anti-repressors, site-specific recombinase, resolvase, and lysis inhibition proteins rIIA and B) replication strategies, these genes are not always definitive indicators of a specific viral replication strategy. Nevertheless, neither Ma-LEF01 nor its relatives were identified as integrated prophages in the 50 bacterial MAGs reconstructed in this study. Ma-LEF01 has a homolog of *nblA* that encodes a phycobilisome degradation protein (red asterisk, Fig. 2A), also described in its relatives (37). The three other Lake Erie *Microcystis* vOTUs were short, sharing only 3–10 kb stretches of homology with other known viruses (Fig. 2B).

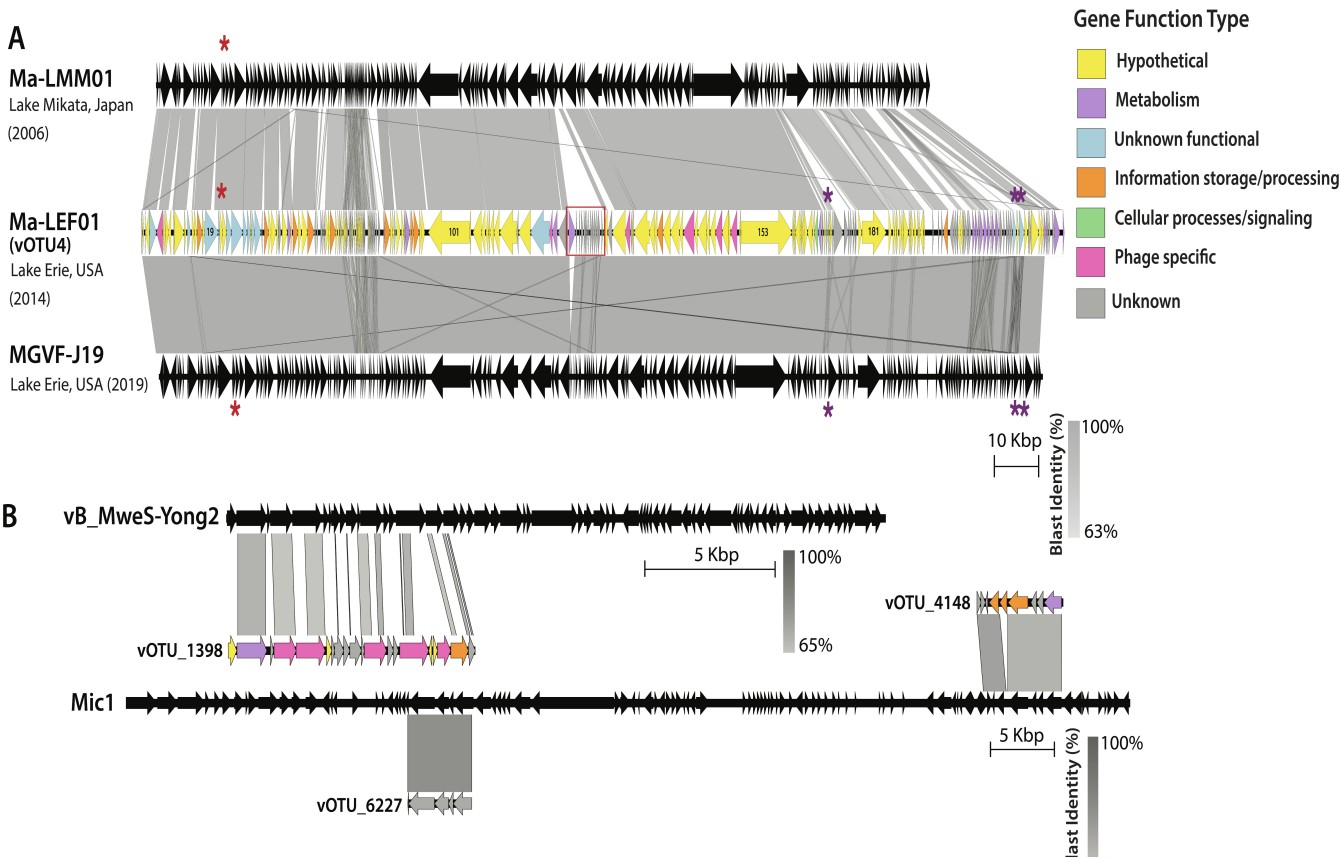

**FIG 2** Lake Erie sampling overview and genome similarity of four closely related *Microcystis* viruses found in Lake Erie. (A) Genome synteny plot of Ma-LMM01, Lake Erie Ma-LEF01, and MVGF-J19 sequences. (B) *Microcystis* virus population synteny plots for non-LMM01/MaMV-DC *Microcystis* virus isolates. Gene color indicated DRAM gene function annotations for Ma-LEF01. "Unknown" is assigned when there is a lack of information, which could arise from no sequence hits. "Unknown functional" is assigned when a sequence has been identified, but its functional role is unidentified or uncharacterized. "Hypothetical" is assigned when the function is not well-characterized or solely predicted computationally. Gray tracks linking portions of the genome represent nucleotide identity. Red box indicates the sequence portion unique to Lake Erie vOTUs. Red asterisk indicates *nblA* gene; purple asterisk indicates pentapeptide repeat proteins.

## Uncharacterized viral OTUs predicted to infect *Microcystis*

Beyond these previously described *Microcystis* viruses, we sought to identify likely infection linkages between the *Microcystis* MAGs and the uncultivated and uncharacterized viruses from the two 2014 Lake Erie cHAB bloom peaks. We used virus-host interaction predictor (VHIP), a machine learning-based tool that predicts putative virus-host pairs with greater than 87% accuracy by leveraging genome-encoded signals of coevolution (59). We found that at 4 August toxic bloom peak, 2,026 virus-host pairs were predicted between 454 vOTUs and 17 bacterial MAGs (9 of which were *Microcystis* MAGs; Table S1) (Fig. 3A). On 29 September non-toxic bloom peak, 1,995 virus-host pairs were predicted between 339 vOTUs and 24 bacterial MAGs (8 *Microcystis* MAGs; Table S1; Fig. 3B; Fig. S4). Ninety-seven percent of viruses were predicted to infect at least one host (Fig. S5). A total of 793 putative *Microcystis* vOTUs were identified at the bloom peaks (Fig. 3A and B). These 793 vOTUs represent those predicted to infect at least one *Microcystis* MAG; if these were also predicted to infect any non-*Microcystis* bacterial hosts, these hosts were also included in the network. vOTUs not predicted to infect *Microcystis* were not included in this study. Notably, only 16 of these *Microcystis*-vOTU connections were identified based on viral sequence matches to *Microcystis* MAG CRISPR spacers (Fig. S6). The putative *Microcystis* vOTUs ranged in sequence length from 10,015

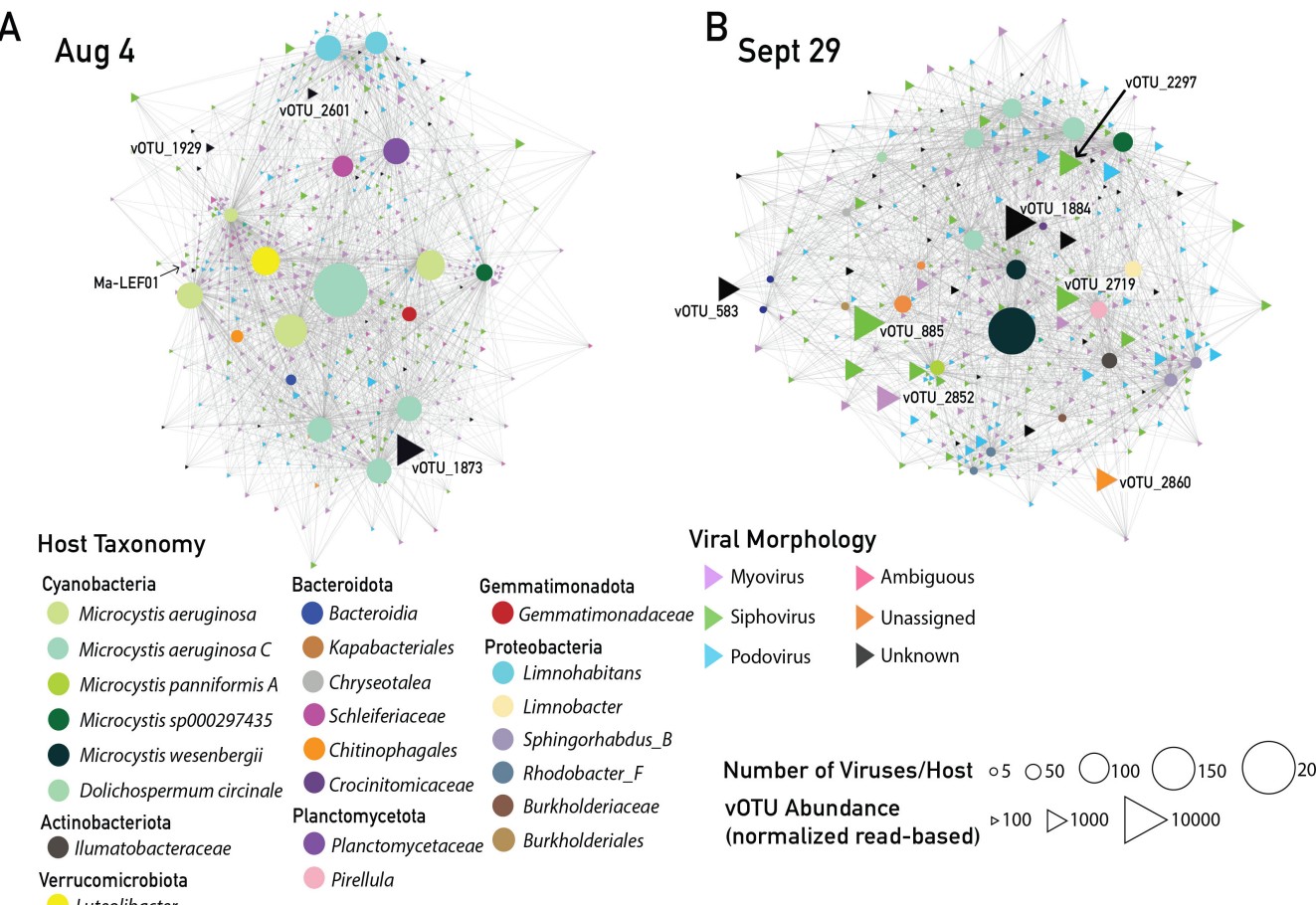

**FIG 3** Networks of predicted interactions between putative *Microcystis* viruses and bacterial host MAGs identified on 4 August and 29 September bloom metagenomes. (A) Predicted virus-host network of 4 August bloom peak. (B) Predicted virus-host network of 29 September bloom peak. Circles are host MAGs; circle size represents the number of viruses predicted to interact with a given host. Triangles are viruses. Node size represents length-adjusted normalized vOTU abundances (see Materials and Methods). Node colors represent assigned morphology; "Unknown" indicates vOTU has no hit in the reference database, "Unassigned" indicates vOTU hits an unassigned virus in the reference database. Only predictions with >93% interaction probability are shown. The highly abundant vOTUs are labeled in each network (identified in Table S6; Fig. 4C).

to 399,821 bp, with an average of 38,916 bp (Fig. S7B), approximately half of the mean genome length reported for bacteriophages in NCBI, which is 68.4 kb.

Abundant vOTUs, which we defined as those recruiting >0.1% of total reads mapped to vOTUs, represented 6.6% and 13.9% of the total *Microcystis* vOTUs on 4 August and 29 September, respectively. The 10 most abundant vOTUs are identified (Fig. 3A and B) and further evaluated in later sections. Most viruses predicted to infect *Microcystis* were present at low abundances at the bloom peaks, especially on 4 August. Notably, Ma-LEF01, relative of Ma-LMM01 that has commonly been used as a biomarker of *Microcystis* virus abundance and infection, is observed on 4 August, but not 29 September, and is not highly abundant at any date.

## Diversity of predicted *Microcystis* vOTUs corresponded with colony formation

### *Microcystis virus alpha diversity was lowest in colony-associated fractions and highest in non-colony fractions*

vOTU abundances, estimated from sequence read recruitment, were used to evaluate alpha and beta diversity trends within vOTUs predicted to infect *Microcystis*. Shannon's evenness was highest in the "non-colony-associated" (viral, 3 µm, and >0.22 µm fractions) and lowest in the "colony-associated" fractions (53 and 100 µm fractions), with the 100 µm fraction least even (Fig. 4A). These differences were significant between the 100 µm fraction and both the >0.22 µm (*P* value = 0.03) and 3 µm fractions (*P* value = 0.04) (Fig. 4A; Table S7).

### *Shifts in predicted Microcystis virus assemblages correlated with sampling fraction and date*

*Microcystis* vOTU assemblage beta diversity, as measured by Bray-Curtis, significantly correlated with filter fraction (Fig. 4B; PERMANOVA $R^2$ = 0.21, *P* value = 0.0001; Table S8) and, to a lesser extent, with sampling date ($R^2$ = 0.06, *P* value = 0.001), but not sample station ($R^2$ = 0.08, *P* value = 0.30). Overall, when variation was visualized in an NMDS ordination (stress = 0.12), *Microcystis* vOTU assemblages partitioned based on colony-associated versus free-living size fractions (Fig. 4B). Of the environmental variables tested, only photoactive radiation was significantly correlated with variation in *Microcystis* vOTU assemblages (Fig. 4B; $R^2$ = 0.06, *P* value = 0.0014; Table S9).

vOTU relative abundances were used to assess shifts in the morphological representation of putative *Microcystis* vOTUs across fractions and bloom peak dates. Of morphologically assigned vOTUs, myoviruses rose to the greatest relative abundance on 4 August, especially in the colony-associated fractions (53 and 100 µm; Fig. 4C). On 29 September, the most abundant myovirus vOTUs belonged to the >0.22 µm non-colony-associated fraction. Putative *Microcystis* vOTUs identified as siphoviruses were also most abundant in the colony-associated fractions. In all cases, shifts in morphological representation between fractions and dates could be attributed to the behavior of a few vOTUs of that class, rather than an overall shift in the entire morphological group (Fig. 4C).

### *High turnover of abundant putative Microcystis virus populations*

To better understand the seasonal fluctuation and host ranges of potentially important putative *Microcystis* vOTUs in the 2014 Lake Erie cHAB, we identified the ten most abundant putative *Microcystis* vOTUs across all samples (Fig. 3A, B and 4D; Table S6). Notably, Ma-LEF01 was absent from this group, indicating its relatively low abundance in the sampled fractions and bloom events. The abundant vOTUs in the cellular fractions were never abundant in the viral fraction (Fig. 4D). The highest relative abundances of abundant vOTUs were typically observed in either the colony-associated or unassociated fractions during the 4 August and 29 September bloom events (Fig. 4D), with only 3 of the 10 vOTUs found to be highly abundant across both dates (Fig. 4D). Of the 10 most abundant vOTUs, 8 were primarily found in the colony-associated 53 or 100 µm fractions. Furthermore, only 4 of the 10 vOTUs could be morphologically classified (Fig. 4D).

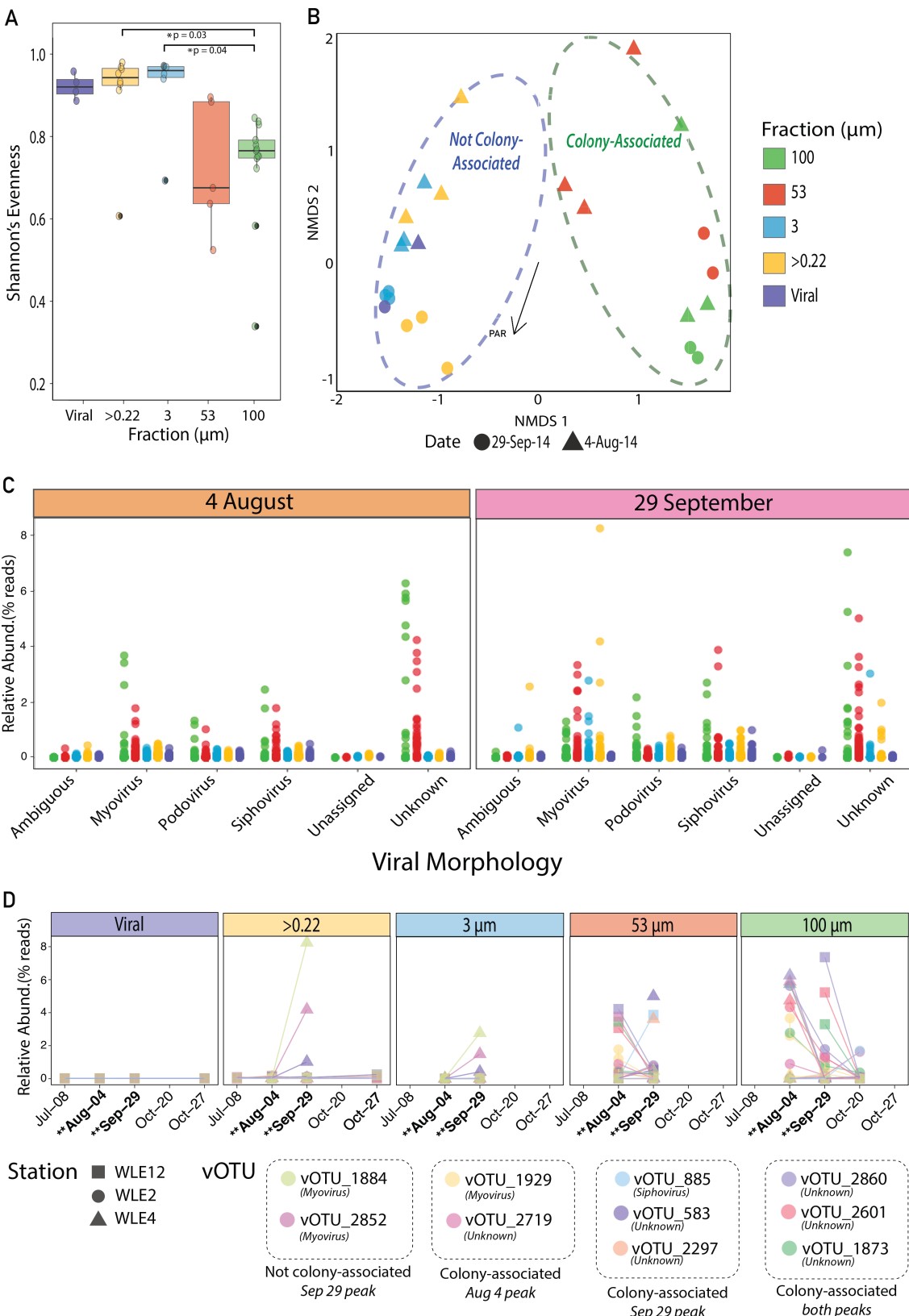

**FIG 4** Putative *Microcystis* virus diversity, variation, and abundant vOTU temporal dynamics through the 2014 Lake Erie bloom season. (A) Diversity of predicted *Microcystis* vOTU assemblage across fractions. (B) NMDS ordination based on Bray-Curtis dissimilarities in distributions of predicted *Microcystis* vOTUs at the cHAB peaks. Point color represents the sampling fraction; shape represents the sampling date. Dashed ovals indicate whether a point belongs to the free-living or

**Fig 4 (Continued)**

colony-associated community. The ordinations are overlaid with the gradient of fit between measured environmental parameters and Bray-Curtis dissimilarities. Vector length of the environmental parameter represents the strength of the correlation with the data variation. (C) Morphological relative abundance distribution of all putative *Microcystis* vOTUs during 4 August and 29 September bloom peaks. Point color represents the sampling fraction. (D) Temporal relative abundance dynamics of the 10 most abundant predicted *Microcystis* vOTUs (Table S6). Point shape represents the sampling station and point color represents a specific putative *Microcystis* virus OTU. Dates of sampled bloom peaks are highlighted in bold with double asterisks.

## Insights on host range and gene flow through *Microcystis* virus-host interaction networks

### Most predicted Microcystis vOTUs had within-genus host ranges, some spanned phyla

To evaluate *Microcystis* vOTU host range and their potential for virus-mediated between-host gene transfer, we identified putative *Microcystis* vOTUs also predicted to infect non-*Microcystis* MAGs. While most shared viruses were shared between *Microcystis* MAGs (Fig. 5A through D), 8 and 16 non-*Microcystis* hosts on 4 August and 29 September, respectively, were predicted to share viruses with *Microcystis* (Fig. 5A and B). VHIP was not designed to distinguish differences at the subspecies level (59), so differences in interaction profiles between highly similar MAGs were not considered in further analyses.

We identified the predicted host range of putative *Microcystis* vOTUs in the Lake Erie bloom as "narrow" (infecting one to two hosts) and "broad" (infecting three or more hosts) using classification criteria from Morimoto et al. (49). Focusing on *Microcystis* hosts alone, we found that 50% of vOTUs were predicted to infect a single host during the bloom peak, and 80% were predicted to infect one or two *Microcystis* MAGs (Fig. 5C). In contrast, when considering all potential hosts, 67% of vOTUs were predicted to infect at least three hosts (Fig. 5D). We found no significant correlations between vOTU host-range breadth and abundance (Table S10).

### Microcystis vOTU metabolic genes can be bloom peak-specific

We identified the metabolic genes carried by putative *Microcystis* viruses to evaluate the potential for virus-mediated gene flow between bacterial populations during the bloom. Most *Microcystis* vOTU genes identified encode proteins with unknown metabolic functions (Fig. 6A; Table S11). Of the AMGs that could be annotated, some were shared at the bloom peaks and some were unique to the peaks on either 4 August or 29 September (Fig. 6A).

Virus-encoded AMGs at both bloom peaks included GDP-Mannose 4,6 dehydratase (potential for complex carbohydrate biosynthesis in cell wall/colony formation, cell communication, and biofilm formation) (60), the 2OG-Fe(II) oxygenase superfamily (involved in phycobiliprotein and secondary metabolite production) (61, 62), and the calcineurin-like phosphoesterase superfamily (phosphate metabolism regulation) (63, 64). On 4 August, enriched virus-encoded genes included photosynthetic reaction center proteins (central to photosynthesis during infection), glutamine synthetase and NAD-dependent epimerase/dehydratase (potential to regulate host nitrogen, amino acid, and energy metabolism) (65), and S-adenosylmethionine decarboxylase (may be involved in cellular responses to nutrient, light, and temperature shifts during infection) (66, 67). On 29 September, the most abundant enriched virus-encoded gene was peptidyl-tRNA hydrolase PTH2 (host protein synthesis regulation) (68). Also abundant was the peroxidase gene *ahpC* (involved in cellular responses to oxidative stress during infection).

### Evaluation of viral morphology diagnostic genes

In order to assess prior reports of specific virus families playing a role in *Microcystis* cHAB progression and demise, we examined the distribution of viral diagnostic genes at both bloom peaks to characterize shifts in the representation of myoviruses, podoviruses, and

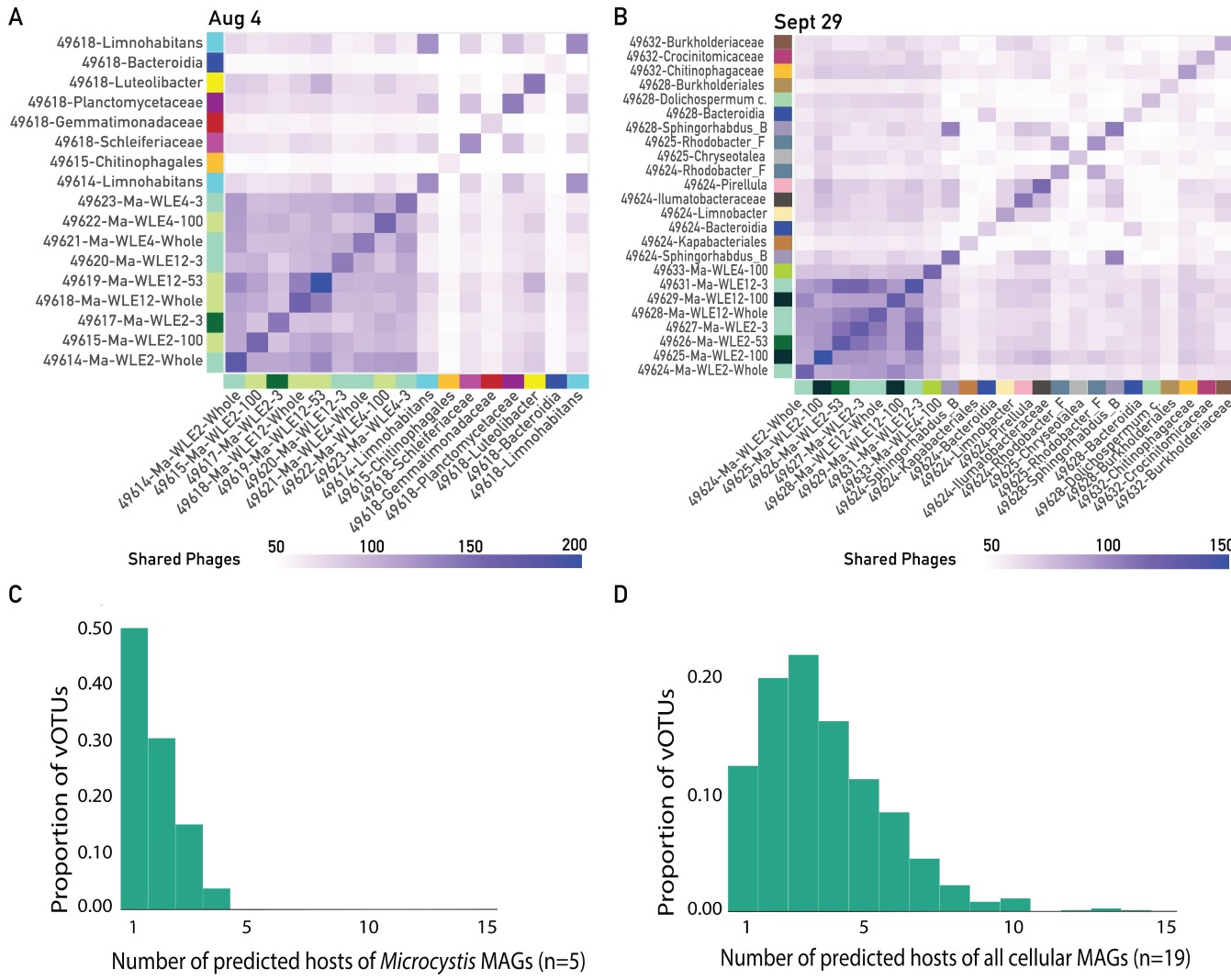

**FIG 5** Heatmaps of shared virus prediction counts between hosts on (A) 4 August and (B) 29 September. Axis colors represent host taxonomy, and heatmap cell color represents the amount of shared phages between any two given hosts. (C) Putative *Microcystis* vOTU host range considering only *Microcystis* hosts. (D) Putative *Microcystis* vOTU host range considering all hosts.

siphoviruses (Fig. 6B; Table S12). Some myovirus diagnostic genes (T4-like terminase large subunit, *gp20* bacteriophage T4-like portal protein) were relatively abundant at both dates. However, other myovirus diagnostic genes (T4 loader gene *gp59*, T4-like virus tail tube protein *gp19*) were distinctly more abundant on 4 August. Similarly, siphovirus diagnostic genes (lambda tail assembly protein, lambda phage tail tape-measure protein, lambda family phage portal protein) showed higher relative abundances during the 4 August bloom peak. In contrast, podovirus diagnostic genes (T7 capsid assembly protein, T7 tail fiber protein, *polA* T7 DNA polymerase, and HK97 putative tail protein *gp10*) were enriched during the 29 September bloom peak.

## DISCUSSION

### Fraction-specific *Microcystis* variation corresponds with virus-host associations and viral diversity through the bloom

This Lake Erie cHAB study revealed fraction-specific population structure of *Microcystis* MAGs. The low evenness of *Microcystis* MAG abundances in the 100 µm fraction suggested the clonal expansion of a few dominant phylotypes, whereby they may

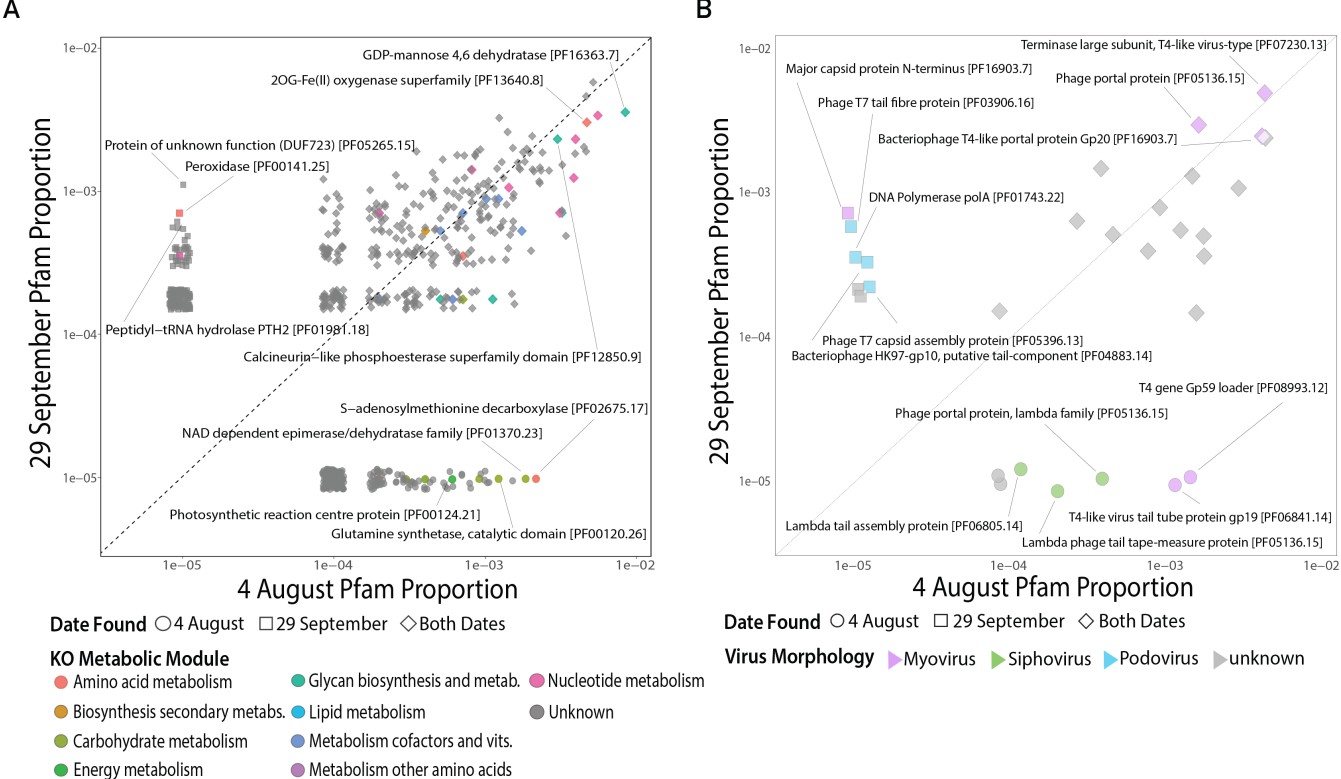

**FIG 6** Putative *Microcystis* virus metabolic genes and viral family diagnostic genes on 4 August and 29 September. (A) Proportion of viral genes encoded by predicted *Microcystis* vOTUs at 4 August bloom peak (*x*-axis) and 29 September bloom peak (*y*-axis). Point color reflects the assigned KEGG Ontology (KO) metabolic module. Points labels are derived from assigned protein family (Pfam) annotations for select functions of interest. (B) Proportion of viral morphology diagnostic genes encoded by predicted *Microcystis* vOTUs at 4 August bloom peak (*x*-axis) and 29 September bloom peak (*y*-axis). The shape indicates which dates viral genes were identified in. Point color reflects viral morphology.

have outcompeted others to form colonies during bloom development. Distinctive phylotypes emerging in the 100 µm fraction at each sampling date (Fig. 1C and D) indicated temporal shifts and successional patterns that may be driven by environmental changes. This pattern aligns with earlier reports of distinct *Microcystis* oligotypes at each bloom peak (69), as well as genotype-level shifts of *Microcystis* at each bloom peak, where a transition from full *mcy* toxin gene cassette to partial *mcy* gene cassette was observed that corresponded with high and low toxin production, respectively (70). This genotype-level shift was hypothesized to be due to reduced ammonium and nitrate levels that may constrain the production of nitrogen-rich microcystin metabolites (70, 71).

Genetic differentiation of *Microcystis* across different size fractions suggested a genome-level response to microenvironmental factors during blooms. The emergence of two genotypic groups within the 100 µm size fraction at bloom peaks may indicate temporal genetic shifts driven by environmental changes or succession dynamics (3). In the principal components analysis of variation among gene clusters by size fraction, the first two components of the *Microcystis* MAG accessory gene clusters offered similar explanatory power (46.8%; Fig. 1G) as when compared to the full complement of *Microcystis* pangenome gene clusters (44.7%; Fig. S2). This suggested that the gene representation in the flexible portion of the *Microcystis* pangenome corresponds with size fractions, though little of this difference is functionally annotated (Table S2). This finding is consistent with a classification framework recently proposed by Cai et al. (72), in which the variable accessory genome reflects the ecological adaptations of *Microcystis*, potentially underlying how *Microcystis* lineages interact with both

biological and physiochemical components of their environments. While fraction-specific adaptations are consistent with niche partitioning observed in cyanobacterial communities (73), fractions also reflect multiple factors, including colony-forming ability and historical fitness advantages that lead certain genotypes to dominate specific fractions at particular times.

These *Microcystis* MAG fraction-specific patterns corresponded with virus-host associations and viral diversity. For instance, the reduced diversity of *Microcystis* MAGs in the 100 µm fraction corresponded with reduced vOTU diversity in that fraction as well (Fig. 1F; Fig. 4A). This pattern, combined with sporadic peaks of putative *Microcystis* vOTUs in the 100 µm fraction (Fig. 4C and D), could be explained by a "Kill the Winner" (KtW) dynamic (74), whereby the dominance of a particular host phylotype selects for a subset of viral types that infect it, reducing overall viral community diversity. In contrast, non-colony-associated *Microcystis* populations likely maintain a more diverse range of viral associations, resulting in higher vOTU diversity within the >0.22 µm fraction.

The observed patterns in viral beta diversity indicated that, as with genotypic variation in *Microcystis* MAGs, changes in viral community composition corresponded with time and size fraction. The correlation between *Microcystis* vOTU diversity and filter fraction and between *Microcystis* vOTU diversity and photosynthetically active radiation (PAR) suggests that light availability is linked to factors influencing *Microcystis* infection dynamics. Given that light intensity and duration impact *Microcystis* growth and physiology in culture (75), these factors likely affect viral interactions *in situ*, as well.

Overall, the observed patterns of *Microcystis* population structure, diversity, and dynamics of *Microcystis* viruses suggested inherent population-level connections between *Microcystis* and their viruses that manifest at the size scale of filter fractions. These findings highlight the importance of disentangling *Microcystis* population dynamics and corresponding *Microcystis* virus-host associations at micron scales. Given the distinct genomic and physiological characteristics that underlie free-living and particle-associated bacterial populations (76, 77), it is likely that important genetic and physicochemical variations occur within microenvironments, such as those created by the formation of colonies. Our data suggests these variations likely correspond with how microbial predator-prey relationships manifest in these blooms. These relationships can have cascading impacts on the ecological and evolutionary pressures *Microcystis* faces during, for example, toxic (early August) and non-toxic (late September) bloom peaks. Tracking the predicted *Microcystis* viral populations at discrete size fractions advances our understanding of how viral predation may influence *Microcystis* population dynamics, which may in turn impact bloom toxicity.

## Implications of virus-host gene exchange for *Microcystis* metabolism and evolution

The *Microcystis* virus-encoded AMGs specific to different bloom peaks may help alleviate metabolic and physiological bottlenecks faced by *Microcystis* in various bloom phases. For instance, bloom-peak-specific photosynthesis genes encoding a photosynthetic reaction center, photosystem II assembly proteins, and cytochrome C were identified and may modulate light-driven energy production for viral replication during high-biomass phases. AMGs encoding for *nblA* (phycobilisome degradation protein), glutamine synthetase, and *nifU* (nitrogenase iron-sulfur cluster scaffold protein) may assist in nitrogen assimilation/reallocation under nutrient limitation, while AMGs encoding S-adenosylmethionine decarboxylase, peroxidase, iron/manganese superoxide dismutase, and molybdopterin oxidoreductase may buffer the host against reactive oxygen species produced during bloom collapse or viral lysis. The bottleneck-relieving function of viral AMGs has been previously proposed (34, 78); this study suggests such a role in the flow of matter and energy within cHABs.

Our analysis of shared phage-host gene clusters suggested the highest levels of virus-mediated gene flow within the *Microcystis* genus (Fig. S8), highlighting the predominance of within-genus phage-host associations. Further, the date-dependent

grouping of shared *Microcystis* virus-host genes (Fig. S8) suggested these associations are dynamic and bloom-stage specific. Such virus-mediated horizontal gene transfer can promote genetic diversity and phenotypic plasticity, enabling *Microcystis* to adapt to varying seasonal conditions. This parallels findings by Hanson et al. (79), who demonstrated that closely related North American coastal cyanophage communities undergo significant temporal variation in host range. They found cyanophage isolates from different seasons varied in their ability to infect specific *Synechococcus* strains, suggesting that virus-host associations are highly dynamic with selection pressures driving host susceptibility over short timescales. Similarly, our findings highlight that Lake Erie viruses are not only top-down population controls on *Microcystis*, but also may shape the metabolic capacities through date-dependent genotypic and phenotypic adaptations.

## Ma-LEF01 genomic features reflect localized and temporal evolution in *Microcystis* viruses

The strain-specific loci of Ma-LEF01 reflect either viral strain diversity arising from evolutionary pressures in different years and locations or strain diversification on spatial (e.g., Lake Erie versus lakes in China and Japan) and temporal (e.g., Lake Erie populations in 2014 versus 2019) scales. The genomic similarity of Ma-LEF01 to MVGF-J19 and Ma-LMM01-like viruses supports the broad geographic distribution of these viruses, while the presence of unique loci indicates localized evolutionary processes. Taken together, the observed genomic features of Ma-LEF01 expand the known diversity of *Microcystis*-infecting viruses and provide insights into population genomic variation across spatial, temporal, and environmental contexts. Despite its historical use as a genetic marker for *Microcystis* viruses, the low abundance of Ma-LEF01 across sampled time points, combined with the low frequency of Ma-LMM01-infected *Microcystis* cells previously observed in Japan—ranging from 0.002% to 1.5% and typically below 0.3% throughout a year-long study (80)—underscores the limitations of relying on single-virus tracking for understanding *Microcystis*-virus dynamics. Our findings support the continued use of a whole community metagenomic approach to more completely capture the viral populations influencing cHAB dynamics.

## *Microcystis* virus host range considerations and its role in shaping cHAB communities

Of the vOTUs predicted to infect *Microcystis*, the majority were classified as "narrow" host-range viruses, similar to the findings of Morimoto et al. (49). However, when expanding the scope to include all potential hosts (not just *Microcystis*), we observed that the majority of those vOTUs (67%) fell into Morimoto's "broad" host-range category, infecting three or more hosts (Fig. 5C and D). In contrast to the Morimoto et al. (49) study, no significant correlation was found between vOTU host range breadth and abundance within *Microcystis* hosts. We attribute this to the variable definitions of broad- and narrow-host range, which emphasizes the importance of explicitly defining host range in viral ecology studies. Different definitions of what is meant by taxonomically "broad" versus "narrow" host ranges lead to varying interpretations of viral success and abundance.

To this point, when we evaluated a wider range of host taxa (i.e., all hosts, not only *Microcystis* hosts), our findings aligned more closely with Morimoto et al. (49) observations that broad-host-range viruses dominated the bloom. Notably, the terms "broad" and "narrow" host range are operational and often defined relative to the scope of a given study. For instance, in the Morimoto study, host diversity was constrained to a single species, whereas our investigation identified host pairs spanning multiple phyla, with shared gene content suggesting potential cross-phyla interactions (Fig. S8). Furthermore, we found GC content to be the most important feature in distinguishing these cross-phyla predictions (Fig. S9). Previous *Microcystis* virus studies have varied in their definitions of broad versus narrow hosts. *Microcystis* viruses Ma-LMM01, Mic1, and

*Microcystis* virus genome fragment MVGF_NODE620 are described as having narrow-host ranges (37, 45, 49), whereas Ma-LMM01 and MVGF_NODE620 are described as being less abundant unless their preferred host strain becomes dominant (49, 81). In contrast, other *Microcystis* viruses exhibit broader host ranges. Though *Microcystis* virus MaMV-DC was initially reported to infect and lyse only a single *M. aeruginosa* strain (FACHB-524) (38), it was later found to infect multiple *Microcystis* species (*M. aeruginosa*, *M. flos-aquae*, and *M. wesenbergii*), functioning more as a genus-level virus (82). Similarly, *Microcystis* virus genome fragment MVGF_NODE331 was found to increase in abundance proportionally to the total *Microcystis* population, regardless of genetic composition (49). The "narrow" versus "broad" host-range distinction reflects complex viral-host interactions. Narrow-host ranges may fit traditional arms-race models, but broad-host ranges can blur taxonomic boundaries, promote cross-phyla gene flow, and demand more nuanced modeling efforts.

Overall, the taxonomic breadth of predicted host associations suggests viruses contribute to the formation, progression, and population structure of the broader cHAB bacterial community, not only *Microcystis* populations. Our cross-phyla interaction predictions are supported by both culture-based freshwater experiments (83) and chromosomal linking methods from deep sea microbial mats (84). Together, this suggests viruses are active architects of microbial community structure that infect hosts not only across the species and genus levels at which most viruses are evaluated (59), but that they likely shape host assemblages and redistribute genetic material cross-phyla, as well.

## Shift in *Microcystis* virus populations hints at shift in infection strategies through bloom phases

Temporal and spatial patterns of viral marker genes offer clues about how *Microcystis* viruses influence and adapt to cHAB phases. Pound et al. (50) observed in a 2014 *Microcystis* bloom in Lake Tai (China) marker genes they associated with myoviruses were more prevalent among early-season host genotypes, while those associated with siphoviruses were more prominent later in the bloom. Similarly, our study found the distribution of marker genes associated with myovirus (T4-like), siphovirus (lambda-like), and podovirus (T7-like) morphotypes shifted across bloom phases. While Pound et al. reported a lytic-to-lysogenic shift in marker gene abundances, our data showed a temporal transition in the prevalence of diagnostic genes associated with T4-like myoviruses (dominant on 4 August) and T7-like podoviruses (more abundant by 29 September), alongside a stable presence of diagnostic genes associated with siphoviruses at each bloom peak. However, viral morphological designation can offer only general clues about whether a virus is more likely to be virulent or temperate; it is not a definitive predictor, especially when relying on sequence data alone. Further, evidence supports that rather than being a simple virulent-temperate dichotomy, virus-microbe interactions likely span a continuum from antagonistic to beneficial and that how they manifest in nature is context-dependent (85). Continued investigation on the role of abiotic factors, suboptimal hosts, multitrophic partnerships, and long-term infections (85) in these blooms will improve our understanding of viral impacts on cHAB microbial community dynamics.

## Considerations of VHIP compared to previously used *Microcystis* virus-host predictions

The virus-host linkages predicted from the VHIP model offered insights into the diversity, specificity, and evolution of *Microcystis* viral associations in the Lake Erie cHAB, and like all methods, rigorous interpretation is prudent. Incomplete viral genome assembly and binning, a common problem in viral metagenomics (36, 86), can lead to an overestimation in the number of virus-host linkages. The mean viral bin length in this study was approximately half that of the mean genome length reported for bacteriophages in

NCBI, indicating the presence of fragmented viral genomes that could lead to artificially inflated linkage counts but does not influence the likelihood of a vOTU-host linkage.

Despite these constraints, VHIP 1.0 predicts interaction pairs with 87.8% accuracy (59) and it addresses limitations of previously used viral host prediction methods, such as marker gene analyses (47, 87), which provides only a narrow view of specific viral populations, and CRISPR-based approaches, which limits the analysis to only the few hosts with detectable CRISPR systems (52). By utilizing multiple genomic features, VHIP captured broader and more diverse relationships, offering higher prediction accuracy and a more inclusive view of viral associations. This approach captured current, historical, and theoretical associations, including those between populations that may not coexist in the same habitat or time and can therefore be interpreted as a summation of potential infection opportunities.

## Outlook

The patterns uncovered in this work highlight the intricate virus-host dynamics at play in a *Microcystis*-dominated cyanobacterial bloom. Additional studies to identify viral gene and protein expression patterns will help to identify relationships between viral activity and community-level metabolite (including toxin) production through the blooms, shedding light on specific viral strategies for manipulating host metabolism, toxin regulation, and avoiding host antiviral strategies within cHABs. Moreover, longitudinal studies encompassing multiple bloom seasons and locations can contribute valuable insights into the temporal and spatial dynamics of putative *Microcystis* vOTUs and host coevolution over long time scales. Understanding connections between the *Microcystis* viruses, the total virus community, environmental parameters, and overall bloom progression is essential to better predict their rise and demise.

## MATERIALS AND METHODS

### Field sampling and collection

Field sites were sampled with the joint NOAA Great Lakes Environmental Research Laboratory/University of Michigan Cooperative Institute for Great Lakes Research weekly sampling program for Lake Erie. In 2014, three sites were sampled bi-monthly in June, then weekly from July through October. Bloom stages were determined by phycocyanin fluorescence and relative abundance. Water chemistry measurements are detailed in Berry et al. (69). Metagenomic data were generated from samples collected from three regularly-sampled stations (WE2, near the mouth of Maumee River, 41°45.743′N, 83°19.874′W; WE4, offshore toward the center of the western basin, 41°49.595′N, 83°11.698′W; and WE12, adjacent to the water intake crib for the city of Toledo, 41°42.535′N, 83°14.989′W). All samples were collected on-station using a peristaltic pump to obtain 20 L of water from 0.1 m below the surface. Water was filtered onto 100 µm polycarbonate filters. This size was used to concentrate *Microcystis* colonies retained on the filter while excluding smaller particles. Previous work has shown that in Lake Erie, the >100 µm assemblage comprised over 90% of all *Microcystis* cells in the water column (88). The filtered water was subsequently passed through a 53 µm and 3 µm polycarbonate filter to collect smaller colonies and large single-celled microbes, including *Microcystis*, whose cell sizes range from 1.7 to 7 µm in diameter (17). Whole water was passed through a 0.22 µm filter to collect the total cellular microbial community; community structure of whole community fractions has been shown to be dominated by single cells (89). To enrich for viruses, 10 g/L iron chloride stock solution (0.966 g $FeCl_3$-$6H_2O$ in 20 mL of 0.02 µm-filtered autoclaved MilliQ-$H_2O$) was added to the <0.22 µm (viral) fraction (90). The flocculant incubated overnight to maximize virus recovery before being passed through a 0.45 µm 142 mm Millipore Express Plus filter and stored at 4°C.

## DNA extraction and sequencing of hosts and viruses

DNA was extracted from samples using the DNeasy Mini Kit (QIAGEN) according to the manufacturer's instructions. Shotgun sequencing of DNA was performed on the Illumina HiSeq platform (2000 PE 100, Illumina, Inc., San Diego, CA, USA) at the University of Michigan DNA Sequencing Core.

## Host assembly, binning, and read mapping

BBDuk (91) was used for read quality trimming, length trimming (anything <100 bp), identifying and removing contaminated sequences against univec, and evaluating denoised reads using FastQC (92). The reads were dereplicated using BBnorm (91). Reads of all 36 samples were assembled on a per-sample basis into contigs using Megahit with metasensitive parameters (93). Following contig assembly, Centrifuge (94) was used to taxonomically classify reads to the species level, or lowest resolution available. Quality-trimmed raw reads were mapped to each individual assembly using bwa (95) with bwa-mem on default settings. SAMtools (96) was used to convert, sort, and index compressed BAM files. Quality-trimmed reads were competitively mapped to MAGs using a pileup shell script provided by BBtools (91). Automated binning was performed on the contigs using Concoct on default settings (97) to generate MAGs. Bins with >50% completeness (completeness statistics inferred from a CheckM [98]), >10% contamination, and <75% strain heterogeneity were manually refined in Anvi'o based on differential coverage and contamination. The Anvi'o platform v2.3.0 (99) was used to manually refine the unique MAGs identified through Concoct and Vizbin (100) by evaluating differential coverage patterns across the samples. Multiple rounds of Anvi'o refinement were performed to curate bins until they passed the aforementioned thresholds (Table S13). To determine host relative abundance of hosts in both the 4 August and 29 September bloom peaks, filtered and trimmed reads were mapped to host MAGs using Bowtie2 (101). Mapped reads to MAGs were summarized with CoverM v.0.6.1 (93), where bins with 70% coverage at 1× read depth were considered present in a given sample. Host MAGs were analyzed for prophage sequences using PHASTEST (102). We used BLAST (103, 104) to determine if any of our putative *Microcystis* vOTUs, classified as prophages by VIBRANT v1.2.1 (105), were detected in host MAGs.

## Viral population identification

CheckV v0.7.0, VIBRANT v1.2.1, VirFinder v1.1, VirSorter v1.0.6, and VirSorter2 v2.1 were used to identify viral contigs according to previously evaluated criteria (36, 105–109). Briefly, a contig was deemed viral if it met any of the following criteria: (i) categorized as category 1 or 2 by VirSorter, (ii) rated high or medium by VIBRANT, (iii) classified as complete, high, or medium by CheckV, (iv) achieved a VirSorter2 score exceeding 0.95, (v) contained at least two hallmark viral genes according to VirSorter2, or (vi) had a VirFinder score above 0.9. Additionally, a contig was considered viral if it was identified as such by at least two of the following conditions: VirSorter categories 3-6, VIBRANT low, CheckV low, VirFinder score between 0.7 and 0.9, and VirSorter2 score between 0.5 and 0.95. Contigs were excluded from the viral list if they (i) lacked viral genes and contained more than one host gene according to CheckV, (ii) had a ratio of host genes to viral genes greater than 3:1 based on CheckV, or (iii) were longer than 50 kb and lacked hallmark viral genes as determined by VirSorter2. Additionally, bacterial genes identified at the edges of proviruses by CheckV, VirSorter2, and VIBRANT were removed. Additionally, only contigs >3 kb were kept from the viral prediction tools and used to identify viral populations. Viruses were then binned using vRhyme default settings to reconstruct a collection of viral bins and high-quality unbinned contigs for population clustering (86). Viral bins and unbinned contigs were clustered (stampede-clustergenomes) (110) according to previously established standards defining viral populations (58). Contigs sharing an average nucleotide identity of 95% across 85% of the contig length were

clustered, and the longest sequence of each cluster was considered the representative for a cluster, referred to as a vOTU.

## Viral community read mapping, quantification, and alpha/beta diversity

Filtered and trimmed reads from the same samples used for host assemblies were quantified using Samtools v1.11 (96). vOTUs within a given sample's assembly were indexed with Bowtie2 using the following parameters to ensure low-quality alignments were filtered out as well as read alignments that did not cover the entire read length: -score-min G, 20, 8 -local (101). Cleaned paired-end reads from each sample were aligned to the indexed vOTU references, producing SAM files of mapped reads. SAM files were filtered to retain only properly paired reads and remove unmapped reads using the following parameters: -hS -f 2F 4. SAM files were converted to BAM format and sorted by genomic coordinates using Samtools. Finally, read counts per vOTU and vOTU relative abundances were extracted from the sorted BAM files using CoverM v.0.6.1 (93), where vOTUs with 0.5 breadth (50% of the genome mapped) were retained. The viral reads for each sample were downsampled by 1,000,000 reads for alpha diversity analyses using seqtk v1-3 (111). Alpha diversity measures were calculated using the vegan v2.5-2 package in R v4.0.2 based on downsampled reads. The transcripts per million (TPM) approach was used to determine the counts and length-based normalized relative abundance for each vOTU and used to calculate the Bray-Curtis distance between samples in R using the vegan package and then NMDS ordination was performed. PERMANOVA using the *adonis* function in vegan was used to test the effects of sampling location, sampling date, sampling fraction as well as effects of environmental parameters on the full viral community structure and metabolic potential. Viral contig information and NCBI accession numbers are in Table S14.

## Viral metabolic potential analyses

KEGG and Pfam databases (112) were accessed to assign vOTU gene metabolic annotations using Distilled and Refined Annotation of Metabolism (DRAM) (112) following the generation of open reading frames (ORFs) using Prodigal (113). Feature-Counts (114) from the Subread package (115) was then used to calculate read coverages of the ORFs generated by DRAM, applying the same 0.5 breadth coverage standard as above. Bray-Curtis distances between samples were calculated using the vegdist function followed by an NMDS ordination with the vegan package in R. Water quality parameters were then applied to a PERMANOVA model to evaluate their effects on the abundance of protein families (Pfams). Parameters were considered significantly correlated with a $P$ value $\leq 0.05$.

## Viral morphological assignment and tracking viral families using diagnostic genes

Morphology of viral populations from the two Lake Erie bloom peaks was estimated using the Phage Taxonomy Tool approach (35). While the ICTV taxonomy has shifted to a genome-based approach, for the purposes of this study, we refer to "myovirus," "siphovirus," and "podovirus" to denote morphological groupings based on diagnostic genes. Using a subset of viral contig DRAM Pfam annotations, we identified the relative abundances of a suite of viral diagnostic genes to assess morphological diversity, specifically for the myoviruses, podoviruses, and siphoviruses. For myoviruses, diagnostic genes included: *g20* (portal protein), *g23* (major capsid protein), *g91* (tail sheath protein), *g43* (T4-specific DNA polymerase), *g101* (putative tail fiber), *g51* (baseplate hub assembly catalyst), and structural genes *g7*, *g10*, and *g12*, which represent essential structural and functional components of T4-like myoviruses. For podoviruses, diagnostic genes included: *polA* (T7-specific DNA polymerase), T7 tail fiber genes, and the T7 capsid assembly genes. For siphoviruses, the diagnostic genes included: hk97 (capsid assembly protein), T5 structural genes, and structural genes for the well-known siphovirus, lambda.

We also included integrase, a gene indicative of lysogenic integration, to examine the composition and replication strategies of these viruses at the bloom peaks. It is important to note, however, that while integrases are commonly found in temperate phages, their presence is not universally indicative of lysogenic integration, nor does their absence preclude a temperate lifestyle. Therefore, conclusions regarding replication strategies based solely on integrase presence should be interpreted with caution.

## Gene cluster analyses

To assess gene conservation and diversity, all *Microcystis* genes from both bloom peaks were converted to gff3 format using Bakta (116), then clustered using Panaroo with default parameters and a 90% cluster threshold, a parameter supported by MAG incompleteness considerations (117). BLAST (103, 104) was used to identify shared phage-host sequences. These shared genes were formatted into gff3 format and clustered with Panaroo, as above, and the presence/absence outputs of these gene clusters at both inter/intra-species and inter/intra-genus levels were evaluated. Inter/intra-phyla comparison was not feasible due to the absence of shared genes between putative *Microcystis* phage and hosts in the same phylum as *Microcystis*.

## Virus-host interaction predictor (VHIP) network

VHIP was used to predict virus-host associations among the binned and high-quality unbinned contigs larger than 10 kb and cellular MAGs. Only prediction scores higher than 0.93 were considered. Networks were plotted using Gephi 0.9.0 (118).

## ACKNOWLEDGMENTS

We thank Michelle Berry for early assistance with viral sample collection and processing and Lizy Michaelson for assistance verifying and publishing protocols in protocols.io. We also thank Derek Smith for his efforts and manual curation of metagenomic host MAGs used for this study. We acknowledge the members of the Duhaime lab for years of constructive feedback on the research and writing that went into this work. We are grateful to the NOAA Great Lakes Environmental Research Lab and the Cooperative Institute for Great Lakes Research logistical team and field crew for allowing us to sample with their HAB monitoring efforts and for assisting in field measurements.

Funding was awarded to the Cooperative Institute for Great Lakes Research (CIGLR) through the NOAA Cooperative Agreement with the University of Michigan (NA17OAR4320152). This CIGLR contribution number is 1268.

This research was funded in part by the University of Michigan Water Center, NOAA Michigan Sea Grant Core Program (awards NA22OAR4170084 and NA18OAR4170102), the National Science Foundation grant 2055455, and the NOAA Omics program awarded via the Cooperative Institute of Great Lake Research (CIGLR) through the NOAA cooperative agreements with the University of Michigan (NA17OAR4320152 and NA22OAR4320150).

## AUTHOR AFFILIATIONS

[1]Department of Ecology and Evolutionary Biology, University of Michigan, Ann Arbor, Michigan, USA
[2]Department of Earth and Environmental Sciences, University of Michigan, Ann Arbor, Michigan, USA

## PRESENT ADDRESS

Bridget Hegarty, Department of Civil and Environmental Engineering, Case Western Reserve University, Cleveland, Ohio, USA

## AUTHOR ORCIDs

A. J. Wing ⬛ http://orcid.org/0009-0008-5056-2550
Bridget Hegarty ⬛ http://orcid.org/0000-0002-3291-4451
Vincent J. Denef ⬛ http://orcid.org/0000-0001-7830-8572
Gregory J. Dick ⬛ http://orcid.org/0000-0001-7666-6288
Melissa B. Duhaime ⬛ http://orcid.org/0000-0001-7884-5087

## FUNDING

| Funder | Grant(s) | Author(s) |
|---|---|---|
| University of Michigan | NA22OAR4170084 | Melissa B. Duhaime |
| Michigan Sea Grant, University of Michigan | NA18OAR4170102 | Melissa B. Duhaime |
| National Science Foundation | 2055455 | Melissa B. Duhaime |
| Cooperative Institute for Great Lakes Research (CIGLR) | NA22OAR4320150, NA17OAR4320152 | Melissa B. Duhaime |

## DATA AVAILABILITY

Read data sets are publicly available in NCBI under SRA numbers: SRX4099271– SRX4099280, SRX4099286–SRX4099288, and SRX4099293–SRX4099300 in BioProject no. PRJNA464361 and SRX24032252–SRX24032266 in BioProject no. PRJNA988094. All metagenomic assemblies used to generate host MAGs and identify viral contigs are available under BioSample numbers: SAMN36000421–SAMN36000456 in BioProject no. PRJNA988094. Host MAGs are available under BioSample numbers: SAMN46387310–SAMN46387359. The scripts used here, including viral identification tools, taxonomy assignment, and relative abundance calculations, are freely available at https://github.com/DuhaimeLab/Tracking_putative_Ma_viruses_and_virus-host_interxns_across_distinct_phases_of_Ma_dominated_bloom.

## ADDITIONAL FILES

The following material is available online.

### Supplemental Material

**Supplemental Figures (mSystems00575-25-s0001.pdf).** Figures S1 to S9.
**Supplemental Tables (mSystems00575-25-s0002.xlsx).** Tables S1 to S14.

### Open Peer Review

**PEER REVIEW HISTORY (review-history.pdf).** An accounting of the reviewer comments and feedback.

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
