## [Reviewer comments · mSystems]

Tracking putative *Microcystis* viruses and virus-host associations across distinct phases of a *Microcystis*-dominated bloom

Anthony Wing, Bridget Hegarty, Eric Bastien, Vincent Deneff, Jacob Evans, Gregory Dick, and Melissa Duhaime

Corresponding Author(s): Melissa Duhaime, University of Michigan

Review Timeline:

Submission Date:	April 24, 2025
Editorial Decision:	June 19, 2025
Revision Received:	July 17, 2025
Accepted:	August 19, 2025

Editor: Jeffrey Blanchard

Reviewer(s): The reviewers have opted to remain anonymous.

Transaction Report:

DOI: <https://doi.org/10.1128/msystems.00575-25>

Re: mSystems00575-25 (Tracking putative *Microcystis* viruses and virus-host associations across distinct phases of a *Microcystis*-dominated bloom)

Dear Dr. Melissa B. Duhaime:

There are additional several questions/comments from the reviewers that need to be addressed, before the manuscript can be accepted. Please return the manuscript within 60 days; if you cannot complete the modification within this time period, please contact me. If you do not wish to modify the manuscript and prefer to submit it to another journal, notify me immediately so that the manuscript may be formally withdrawn from consideration by mSystems.

Revision Guidelines

Sincerely,
Jeffrey Blanchard
Editor
mSystems

Reviewer #1 (Comments for the Author):

I appreciate the authors' thoughtful responses and revisions to this manuscript. I think it is an important piece of work for the cHAB community. In general, most of my comments have been adequately addressed, and I think the new structure (split Results/Discussion) is much easier to follow. It is clear a lot of effort went into the revisions.

I have a few relatively minor comments:

1. Gene clustering analysis (Fig 1G, lines 144-147, 317-321: I think the clustering analysis needs a bit of clarification. How were "core" genes defined? If their presence/absence is meaningful (as implied by Fig 1G), then are they actually core genes? My understanding is that core genes are universally shared among the group (though not always universally detected, due to incomplete MAGs). The methods describe a 90% clustering cutoff, in which case some core protein families could conceivably be split into different sequence clusters, and maybe that's the presence/absence that's being plotted in 1G? In that case, it really is more a reflection of sequence similarity (i.e. another proxy for ANI) than for gene content.
2. I still wonder what genes distinguish the MAGs from different size fractions... though I recognize that maybe the authors don't want to delve into that here because it's a bigger question. But maybe a sentence or two would be nice.
3. Network connections (lines 178-182): just to clarify -- are the 793 vOTUs here limited to only those that are associated with *Microcystis* (and sometimes another bacterium as well)? Since other bacteria are shown, I was wondering about vOTUs linked to these other bacteria -- maybe just a sentence stating that there were other vOTUs discovered that did not associate with *Microcystis* (if this is true), but they are not discussed here?
4. line 224-5 incomplete sentence
5. line 239-240: at different sites? "peak" implies singularity, so it took me a minute to understand how they could peak at two times.
6. Fig 6B: maybe color the points by virus family instead of date, since date is shown on the axes? This would make it clear whether the 3 families are more/less abundant on the two dates.

Reviewer #2 (Comments for the Author):

The authors have undertaken significant work to revise this MS. It is easier to follow now that the results and discussion are separated, and the grammar is improved. The introduction has been revised to provide a broader context for the work, situating it within the global context and the work would be of interest to the broad readership of mSystems.

There are a couple of virus taxonomy specific issues that need to be addressed. The first is simple - the name of the host is not italicized when used to describe the virus (*Microcystis virus*), so this should be edited throughout. The second is more difficult and seems to have been introduced with additional data exploration after the first submission.

ICTV revised the taxonomical approach to bacteriophage in 2022. The taxonomy of phage has moved to a genome-based approach, and names based on morphology are no longer accepted, meaning, the names Myoviridae, Siphoviridae, and Podoviridae are no longer valid taxonomic groups. The organisms in these groups are now all reassigned to the class Caudoviricetes.

In this MS, the taxonomic assignment of viral populations was estimated using Phage Taxonomy Tool, and in this case specific gene markers were chosen to differentiate between these old morphological groups of phages. While there are clearly differences in abundance, host-virus pairs, etc., between the different morphological groups, the current grouping into defunct groups is problematic. I'm not sure what is the best approach, but suggest the analysis could be left as is, and the names of the groups be revised to myovirus, podovirus and siphovirus to indicate they are morphological and not taxonomical groups.

There are also a few issues around the prediction of viral replication strategy. Around line 164 it is suggested that tail sheaths are indicative of lytic phages, but temperate phages can switch between lytic and lysogenic strategies, so they also have tail sheath proteins. Likewise, it is suggested that only temperate phages will have lysis inhibition proteins, but lytic phages can also have lysis inhibition proteins in order to delay/regulate host cell lysis. These genes are therefore not good indicators of viral replication strategy.

The section around line 470 attempts to suggest viral replication strategy based on the old morphological groupings of phage. I would suggest this is inappropriate and should be avoided. Line 637-639 indicates that integrase was included in the PTT taxonomic analysis as an indicator of lysogenic integration. While most temperate phage analyzed to date have integrases, some do not, so conclusions based on this analysis should be approached with caution.

[revised manuscript text omitted]

genes was considered, 31.1% of the variation in core gene clusters could be explained
by fraction, with the 100- μm MAGs forming a distinct grouping along principle
component 1 (Fig. 1G).

Identification of *Microcystis* viruses in the 2014 Lake Erie bloom

**Globally-distributed *Microcystis* viruses identified through sequence homology**

From the total pool of contigs, 27,086 viral contigs >3 kb in length were identified, which
were then binned and clustered into 15,461 non-redundant viral operational taxonomic
units (vOTUs), approximating viral species based on currently established thresholds of

no italics when part of virus name

95% ANI across 85% of the contig length (Roux et al., 2019). None of these viruses
 were integrated as prophages in the *Microcystis* MAGs. We identified four putative
 *Microcystis* vOTUs (vOTU_4, vOTU_1398, vOTU_4148, vOTU_6227) with a high
 degree of similarity to four *Microcystis* virus isolates (Fig. 2; SI Table 3). The
 representative virus of vOTU4, renamed here as Ma-LEF01 (*Microcystis aeruginosa*
 Lake Erie Fukuivirus-01), had high similarity (>99% shared average nucleotide identity,
 ANI, across 95% of its genome) to a viral contig, MVGF-J19, that was previously
 assembled from a 2019 Lake Erie cHAB metagenome (SI Fig. 3).⁶¹ MVGF-J19 and Ma-
 LEF01 were also highly similar to *Microcystis* isolates MaMV-DC and Ma-LMM01 (SI
 Fig. 3). All seven of the 243 predicted genes that were unique to Ma-LEF01 were of
 unknown function (SI Table 4). While Ma-LEF01 has genes characteristic of both lytic
 (e.g., viral tail sheath) and lysogenic (e.g., putative phage anti-repressors, site-specific
 recombinase, resolvase, lysis inhibition proteins rIIA and B) replication strategies,
 neither Ma-LEF01 nor its relatives were identified as integrated prophages in the 50
 bacterial MAGs reconstructed in this study. Ma-LEF01 has a homologue of nbIA that
 encodes a phycobilisome degradation protein (red asterisk, Fig. 2A), also described in
 its relatives.¹¹² The three other Lake Erie *Microcystis* vOTUs were short, sharing only 3-
 10 kb stretches of homology with other known viruses (Fig. 2B).

temperate viruses have these too.

lytic viruses also have.

lysis inhibition proteins

**Uncharacterized Viral OTUs predicted to infect *Microcystis***

Beyond these previously described *Microcystis* viruses, we sought to identify likely
 infection linkages between the *Microcystis* MAGs and the uncultivated and
 uncharacterized viruses from the two 2014 Lake Erie cHAB bloom peaks. We used
 Virus-Host Interaction Predictor (VHIP), a machine learning-based tool that predicts

putative virus-host pairs with greater than 87% accuracy by leveraging genome-
encoded signals of coevolution.⁷ We found that at the August 4 toxic bloom peak, 2,026
virus-host pairs were predicted between 454 vOTUs and 17 bacterial MAGs (9 of which
were *Microcystis* MAGs; SI Table 1) (Fig. 3A). On the September 29 non-toxic bloom
peak, 1,995 virus-host pairs were predicted between 339 vOTUs and 24 bacterial MAGs
(8 *Microcystis* MAGs; SI Table 1; Fig. 3B; SI Fig. 4). 97% of viruses were predicted to
infect at least one host (SI Fig. 5). A total of 793 putative *Microcystis* vOTUs were
identified at the bloom peaks (Fig. 3A-B). Notably, only 16 of these *Microcystis* vOTU
connections were identified based on viral sequence matches to *Microcystis* MAG
CRISPR spacers (SI Fig. 6). The putative *Microcystis* vOTUs ranged in sequence length
from 10,015 bp to 399,821 bp, with an average of 38,916 bp (SI Fig. 7B), approximately
half of the mean genome length reported for bacteriophages in NCBI, which is 68.4 kb.

Abundant vOTUs, which we defined as those recruiting >0.1% of total reads
mapped to vOTUs, represented 6.6% and 13.9% of the total *Microcystis* vOTUs on 4
Aug and 29 Sept, respectively. The 10 most abundant vOTUs are identified (Fig. 3A-B)
and further evaluated in later sections. Most viruses predicted to infect *Microcystis* were
present at low abundances at the bloom peaks, especially on August 4. Notably, Ma-
LEF01, relative of Ma-LMM01 that has commonly been used as a biomarker of
*Microcystis* virus abundance and infection, is observed on Aug 4, but not Sept 29, and
is not highly abundant at any date.

Diversity of predicted *Microcystis* vOTUs corresponded with colony formation
***Microcystis* virus alpha diversity lowest in colony-associated fractions, highest in**
**non-colony fractions**

vOTU abundances, estimated from sequence read recruitment, were used to evaluate
alpha and beta diversity trends within vOTUs predicted to infect *Microcystis*. Shannon's
evenness was highest in the 'not colony-associated' (viral, 3- μm , and $>0.22\text{-}\mu\text{m}$
fractions) and lowest in the 'colony-associated' fractions (53 and 100- μm fractions), with
the 100- μm fraction the least even (Fig. 4A). These differences were significant between
100- μm fraction and both the $>0.22\text{-}\mu\text{m}$ (p-value=0.03) and 3- μm fractions (p-
value=0.04) (Fig. 4A; SI Table 6).

**Shifts in predicted *Microcystis* virus assemblages correlated with sampling**
**fraction and date**

*Microcystis* vOTU assemblage beta diversity, as measured by Bray Curtis, significantly
correlated with filter fraction (Fig. 4B; PERMANOVA $R^2=0.21$, p-value=0.0001; SI Table
7) and, to a lesser extent, with sampling date ($R^2=0.06$, p-value = 0.001), but not
sample station ($R^2=0.08$, p-value=0.30). Overall, when variation was visualized in an
NMDS ordination (stress=0.12), *Microcystis* vOTU assemblages partitioned based on
colony-associated versus free-living size fractions (Fig. 4B). Of the environmental
variables tested, only photoactive radiation (PAR) was significantly correlated with
variation in *Microcystis* vOTU assemblages (Fig. 4B; $R^2=0.06$, p-value=0.0014; SI Table
8).

vOTU relative abundances were used to assess shifts in the taxonomic
representation of putative *Microcystis* vOTUs across fractions and bloom peak dates. Of

taxonomically assigned vOTUs, members of the Myoviridae rose to the greatest relative
abundance on August 4, especially in the colony-associated fractions (53- μ m and 100-
μ m; Fig. 4C). While on September 29 the most abundant Myoviridae vOTUs belonged
to the >0.22- μ m not-colony-associated fraction. Putative *Microcystis* vOTUs assigned to
the Siphoviridae family were also most abundant in the colony-associated fractions. In
all cases, shifts in taxonomic representation between fractions and dates could be
attributed to the behavior of a few vOTUs of that class, rather than an overall shift in the
entire taxonomic group (Fig. 4C).

not a valid taxonomic group according to ICTC 2022¹⁰

High turnover of abundant putative *Microcystis* virus populations

To better understand the seasonal fluctuation and host ranges of potentially important
putative *Microcystis* vOTUs in the 2014 Lake Erie cHAB, we identified the ten most
abundant putative *Microcystis* vOTUs across all samples (Fig. 3A-B, Fig. 4D, SI Table
5). Notably, Ma-LEF01 was absent from this group, indicating its relatively low
abundance in the sampled fractions and bloom events. The abundant vOTUs in the
cellular fractions were never abundant in the viral fraction (Fig. 4D). Abundant vOTUs
generally peaked in either the colony-associated or unassociated fractions during bloom
events on Aug 4 or Sept 29 (Fig. 4D), with only three of the 10 vOTUs peaking at both
dates (Fig. 4D). Of the 10 most abundant vOTUs, eight were primarily found in the
colony-associated 53 or 100- μ m fractions. Furthermore, only four of the 10 vOTUs could
be taxonomically classified (Fig. 4D).

names are outdated.
(Sipho / Myo viridae)
- genome-based name is
Caudoviricetes

[revised manuscript text omitted]

Genetic differentiation of *Microcystis* across different size fractions suggested a
genome-level response to microenvironmental factors during blooms. The emergence
of two genotypic groups within the 100- μ m size fraction at bloom peaks may indicate
temporal genetic shifts driven by environmental changes or succession dynamics.⁷⁴ In
the size-fraction principal component analysis, *Microcystis* MAG core gene clusters
offered greater explanatory power than the full complement of genes, implying that the
relatively stable, shared portion of the *Microcystis* genome correlates more closely with

[revised manuscript text omitted]

SRX4099280, SRX4099286 to SRX4099288, and SRX4099293 to SRX4099300 in
BioProject no. PRJNA464361 and SRX24032252 to SRX24032266 in BioProject no.
PRJNA988094. All metagenomic assemblies used to generate host MAGs and identify
viral contigs are available under BioSample numbers: SAMN36000421 to
SAMN36000456 in BioProject no. PRJNA988094. Host MAGs are available under
BioSample numbers: SAMN46387310 to SAMN46387359. The scripts used here,
including viral identification tools, taxonomy assignment, relative abundance
calculations, are freely available at
https://github.com/DuhaimeLab/Tracking_putative_Ma_viruses_and_virus-
[host_interxns_across_distinct_phases_of_Ma_dominated_bloom](https://github.com/DuhaimeLab/Tracking_putative_Ma_viruses_and_virus-host_interxns_across_distinct_phases_of_Ma_dominated_bloom)

Competing interests

The authors declare that they have no competing interests.

Funding

This research was funded in part by the University of Michigan Water Center, NOAA
Michigan Sea Grant Core Program awards NA22OAR4170084, and
NA18OAR4170102, the National Science Foundation grant 2055455, and the NOAA
Omics program awarded via the Cooperative Institute of Great Lake Research (CIGLR)

through the NOAA cooperative agreements with the University of Michigan
(NA17OAR4320152 and NA22OAR4320150).

Acknowledgements

We would like to thank Michelle Berry for early assistance with viral sample collection
and processing and Lizy Michaelson for assistance verifying and publishing protocols in
protocols.io. We would also like to thank Derek Smith for his efforts and manual curation
of metagenomic host MAGs used for this study. We acknowledge the members of the
Duhaime Lab for years of constructive feedback on the research and writing that went
into this work. We are grateful to the NOAA Great Lakes Environmental Research Lab
and the Cooperative Institute for Great Lakes Research logistical team and field crew
for allowing us to sample with their HAB monitoring efforts and for assisting in field
measurements. Funding was awarded to the Cooperative Institute for Great Lakes
Research (CIGLR) through the NOAA Cooperative Agreement with the University of
Michigan (NA17OAR4320152). This CIGLR contribution number is #####.

References

- 1. Alexova R, Fujii M, Birch D, Cheng J, Waite TD, Ferrari BC, Neilan BA. Iron
uptake and toxin synthesis in the bloom-forming *Microcystis aeruginosa* under
iron limitation. *Environ Microbiol.* 2011 Apr;13(4):1064-77.
<https://pubmed.ncbi.nlm.nih.gov/21251177/>
- 2. Alneberg, J., Bjarnason, B. S., de Bruijn, I., Schirmer, M., Quick, J., Ijaz, U. Z.,
Loman, N. J., Andersson, A. F., & Quince, C. (2013). *CONCOCT: Clustering*

- *cONtigs on COverage and ComposiTion* (arXiv:1312.4038). arXiv.
<https://doi.org/10.48550/arXiv.1312.4038>
- 3. Altschul, SF, Gish, W, Miller, W, Myers, EW, Lipman, DJ. Basic local alignment
search tool. *J Mol Biol.* 1990 Oct 5;215(3):403-10. [https://doi.org/10.1016/S0022-
2836\(05\)80360-2](https://doi.org/10.1016/S0022-2836(05)80360-2)
- 4. Anantharaman, K., Duhaime, M. B., Breier, J. A., Wendt, K. A., Toner, B. M., &
Dick, G. J. (2014). Sulfur Oxidation Genes in Diverse Deep-Sea Viruses.
*Science*, 344(6185), 757–760. <https://doi.org/10.1126/science.1252229>
- 5. Andrews, S. (2010) FastQC: A Quality Control Tool for High Throughput
Sequence Data.
- 6. Bastian, M., Heymann, S., & Jacomy, M. (2009). Gephi: An Open Source
Software for Exploring and Manipulating Networks. *Proceedings of the
International AAAI Conference on Web and Social Media*, 3(1), 361-362.
<https://doi.org/10.1609/icwsm.v3i1.13937>
- 7. Bastien, E. G., Cable, R. N., Zaman, L., Batterbee, C., Wing, A. J., & Duhaime,
711 M. B. (2024). *Virus-host interactions predictor (VHIP): Machine learning
approach to resolve microbial virus-host interaction networks* (p.
2023.11.03.565433). *PLOS Computational Biology*.
<https://doi.org/10.1371/journal.pcbi.1011649>
- 8. Berry, M. A., Davis, T. W., Cory, R. M., Duhaime, M. B., Johengen, T. H., Kling,
G. W., Marino, J. A., Den Uyl, P. A., Gossiaux, D., Dick, G. J., & Deneff, V. J.
(2017a). *Cyanobacterial harmful algal blooms are a biological disturbance to*

- *Western Lake Erie bacterial communities.* <https://doi.org/10.1111/1462->
[2920.13640](https://doi.org/10.1111/1462-2920.13640)
- 9. Berry, M. A., White, J. D., Davis, T. W., Jain, S., Johengen, T. H., Dick, G. J.,
Sarnelle, O., & Deneff, V. J. (2017b). Are Oligotypes Meaningful Ecological and
Phylogenetic Units? A Case Study of *Microcystis* in Freshwater Lakes. *Frontiers*
*in Microbiology*, 08, 365–365. <https://doi.org/10.3389/fmicb.2017.00365>
- 10. Bolay, P., Muro-Pastor, M. I., Florencio, F. J., & Klähn, S. (2018). The Distinctive
Regulation of Cyanobacterial Glutamine Synthetase. *Life (Basel, Switzerland)*,
8(4), 52. <https://doi.org/10.3390/life8040052>
- 11. Breitbart, M. (2012). Marine viruses: Truth or dare. *Annual Review of Marine*
*Science*, 4, 425–448. <https://doi.org/10.1146/annurev-marine-120709-142805>
- 12. Burstein, D., Sun, C. L., Brown, C. T., Sharon, I., Anantharaman, K., Probst, A.
730 J., Thomas, B. C., & Banfield, J. F. (2016). Major bacterial lineages are
731 essentially devoid of CRISPR-Cas viral defence systems. *Nature*
*Communications*, 7(1), Article 1. <https://doi.org/10.1038/ncomms10613>
- 13. Bushnell, B. (2014). BBTools software package. <http://bbtools.jgi.doe.gov>
- 14. Cai, H., et al. ,*Microcystis* pangenome reveals cryptic diversity within and across
morphospecies. *Sci. Adv.* 9, eadd3783(2023)
<https://www.science.org/doi/10.1126/sciadv.add3783>
- 15. Cai, R., Li, D., Lin, W., Qin, W., Pan, L., Wang, F., Qian, M., Liu, W., Zhou, Q.,
Zhou, C., & Tong, Y. (2022). Genome sequence of the novel freshwater
*Microcystis* cyanophage Mwe-Yong1112-1. *Archives of Virology*, 167, 1–6.
<https://doi.org/10.1007/s00705-022-05542-3>

- 16. Chaffin, J. D., Bridgeman, T. B., Heckathorn, S. A., & Mishra, S. (2011).
Assessment of *Microcystis* growth rate potential and nutrient status across a
trophic gradient in western Lake Erie. *Journal of Great Lakes Research*, 37(1),
92–100. <https://doi.org/10.1016/j.jglr.2010.11.016>
- 17. Cook, KV, Li C, Cai H, Krumholz LR, Hambright KD, Paerl HW, Steffen MM,
Wilson AE, Burford MA, Grossart HP, Hamilton DP, Jiang H, Sukenik A, Latour
D, Meyer EI, Padisák J, Qin B, Zamor RM, Zhu G. The global *Microcystis*
interactome. *Limnol Oceanogr*. 2020 Jan;65(Suppl 1):S194-S207. doi:
10.1002/lno.11361. Epub 2019 Nov 19. PMID: 32051648; PMCID: PMC7003799.
- 18. Correa, A. M. S., Howard-Varona, C., Coy, S. R., Buchan, A., Sullivan, M. B., &
Weitz, J. S. (2021). Revisiting the rules of life for viruses of microorganisms.
*Nature Reviews Microbiology*, 19(8), Article 8. [https://doi.org/10.1038/s41579-](https://doi.org/10.1038/s41579-021-00530-x)
[021-00530-x](https://doi.org/10.1038/s41579-021-00530-x)
- 19. Cory, R. M., Davis, T. W., Dick, G. J., Johengen, T., Denef, V. J., Berry, M. A.,
Page, S. E., Watson, S. B., Yuhas, K., & Kling, G. W. (2016). Seasonal
Dynamics in Dissolved Organic Matter, Hydrogen Peroxide, and Cyanobacterial
Blooms in Lake Erie. *Frontiers in Marine Science*, 3.
<https://www.frontiersin.org/articles/10.3389/fmars.2016.00054>
- 20. Dziallas C, Grossart H-P (2011) Increasing Oxygen Radicals and Water
Temperature Select for Toxic *Microcystis* sp. *PLoS ONE* 6(9): e25569.
<https://doi.org/10.1371/journal.pone.0025569>

- 21. Enav, H., Kirzner, S., Lindell, D., Mandel-Gutfreund, Y., & Béjà, O. (2018).
Adaptation to sub-optimal hosts is a driver of viral diversification in the ocean.
*Nature Communications*, 9, 4698. <https://doi.org/10.1038/s41467-018-07164-3>
- 22. Eren, A. M., Kiefl, E., Shaiber, A., Veseli, I., Miller, S. E., Schechter, M. S., Fink,
I., Pan, J. N., Yousef, M., Fogarty, E. C., Trigodet, F., Watson, A. R., Esen, Ö. C.,
Moore, R. M., Clayssen, Q., Lee, M. D., Kivenson, V., Graham, E. D., Merrill, B.
D., ... Willis, A. D. (2021). Community-led, integrated, reproducible multi-omics
with anvio. *Nature Microbiology*, 6(1), Article 1. [https://doi.org/10.1038/s41564-](https://doi.org/10.1038/s41564-020-00834-3)
[020-00834-3](https://doi.org/10.1038/s41564-020-00834-3)
- 23. Fontana S, Thomas MK, Reyes M, Pomati F. Light limitation increases
multidimensional trait evenness in phytoplankton populations. *ISME J*. 2019
May;13(5):1159-1167. doi: 10.1038/s41396-018-0320-9. Epub 2019 Jan 7.
PMID: 30617295; PMCID: PMC6474219.
- 24. Fuhrman. (1999). *Marine viruses and their biogeochemical and ecological effects*
| *Nature*. <https://www.nature.com/articles/21119>
- 25. Garcia-Villegas, M., De La Vega, M., Galindo, J., Segura, M., Buckingham, R.,
Guarneros, G. (1991). *Peptidyl-tRNA hydrolase is involved in lambda inhibition of*
*host protein synthesis*. Retrieved January 10, 2024, from
<https://www.embopress.org/doi/epdf/10.1002/j.1460-2075.1991.tb04919.x>
- 26. Guo, J., Bolduc, B., Zayed, A. A., Varsani, A., Dominguez-Huerta, G., Delmont,
782 T. O., Pratama, A. A., Gazitúa, M. C., Vik, D., Sullivan, M. B., & Roux, S. (2021).
VirSorter2: A multi-classifier, expert-guided approach to detect diverse DNA and
RNA viruses. *Microbiome*, 9(1), 37. <https://doi.org/10.1186/s40168-020-00990-y>

- 27. Hanson CA, Marston MF, Martiny JB. Biogeographic Variation in Host Range
Phenotypes and Taxonomic Composition of Marine Cyanophage Isolates. *Front*
*Microbiol.* 2016 Jun 24;7:983. <https://doi.org/10.3389/fmicb.2016.00983>
- 28. Harke, M. J., Steffen, M. M., Gobler, C. J., Otten, T. G., Wilhelm, S. W., Wood, S.
789 A., & Paerl, H. W. (2016). A review of the global ecology, genomics, and
790 biogeography of the toxic cyanobacterium, *Microcystis* spp. *Harmful Algae*, 54,
4–20. <https://doi.org/10.1016/j.hal.2015.12.007>
- 29. Hegarty B, Riddell V J, Bastien E, Langenfeld K, Lindback M, Saini JS, Wing A,
Zhang J, Duhaime M. 2024. Benchmarking informatics approaches for virus
discovery: caution is needed when combining in silico identification methods.
*mSystems* 9:e01105-23. <https://doi.org/10.1128/msystems.01105-23>
- 30. Herr, C. Q., & Hausinger, R. P. (2018). Amazing Diversity in Biochemical Roles
of Fe(II)/2-Oxoglutarate Oxygenases. *Trends in Biochemical Sciences*, 43(7),
517–532. <https://doi.org/10.1016/j.tibs.2018.04.002>
- 31. Howard-Varona, C., Lindback, M. M., Bastien, G. E., Solonenko, N., Zayed, A.
800 A., Jang, H., Andreopoulos, B., Brewer, H. M., Glavina del Rio, T., Adkins, J. N.,
Paul, S., Sullivan, M. B., & Duhaime, M. B. (2020). Phage-specific metabolic
reprogramming of virocells. *The ISME Journal*, 14(4), Article 4.
<https://doi.org/10.1038/s41396-019-0580-z>
- 32. Huisman, J., Codd, G. A., Paerl, H. W., Ibelings, B. W., Verspagen, J. M. H., &
Visser, P. M. (2018). Cyanobacterial blooms. *Nature Reviews Microbiology*,
16(8), 471–483. <https://doi.org/10.1038/s41579-018-0040-1>

- 33. Hurwitz, B. L., Hallam, S. J., & Sullivan, M. B. (2013). Metabolic reprogramming
by viruses in the sunlit and dark ocean. *Genome Biology*, 14(11), R123.
<https://doi.org/10.1186/gb-2013-14-11-r123>
- 34. Hwang, Y., Roux, S., Coclet, C., Krause, S. J. E., & Girguis, P. R. (2023). Viruses
interact with hosts that span distantly related microbial domains in dense
hydrothermal mats. *Nature Microbiology*, 8(5), Article 5.
<https://doi.org/10.1038/s41564-023-01347-5>
- 35. Hyatt, D., Chen, G.-L., LoCascio, P. F., Land, M. L., Larimer, F. W., & Hauser, L.
815 J. (2010). Prodigal: Prokaryotic gene recognition and translation initiation site
identification. *BMC Bioinformatics*, 11(1), 119. [https://doi.org/10.1186/1471-2105-](https://doi.org/10.1186/1471-2105-11-119)
[11-119](https://doi.org/10.1186/1471-2105-11-119)
- 36. Ignacio-Espinoza, J.C. and Sullivan, M.B. (2012), Phylogenomics of T4
cyanophages: lateral gene transfer in the 'core' and origins of host genes.
*Environmental Microbiology*, 14: 2113-2126. [https://doi.org/10.1111/j.1462-](https://doi.org/10.1111/j.1462-2920.2012.02704.x)
[2920.2012.02704.x](https://doi.org/10.1111/j.1462-2920.2012.02704.x)
- 37. Jantaro, S., Mäenpää, P., Mulo, P., & Incharoensakdi, A. (2003). Content and
biosynthesis of polyamines in salt and osmotically stressed cells of
*Synechocystis* sp. PCC 6803. *FEMS Microbiology Letters*, 228(1), 129–135.
[https://doi.org/10.1016/S0378-1097\(03\)00747-X](https://doi.org/10.1016/S0378-1097(03)00747-X)
- 38. Jia, B., Jia, X., Kim, K. H., & Jeon, C. O. (2017). Integrative view of 2-
oxoglutarate/Fe(II)-dependent oxygenase diversity and functions in bacteria.
*Biochimica et Biophysica Acta (BBA) - General Subjects*, 1861(2), 323–334.
<https://doi.org/10.1016/j.bbagen.2016.12.001>

- 39. Jiang, X., Ha, C., Lee, S., Kwon, J., Cho, H., Gorham, T., & Lee, J. (2019).
Characterization of Cyanophages in Lake Erie: Interaction Mechanisms and
Structural Damage of Toxic Cyanobacteria. *Toxins*, 11(8), 444.
<https://doi.org/10.3390/toxins11080444>
- 40. Kehr, J.-C., & Dittmann, E. (2015). Biosynthesis and function of extracellular
glycans in cyanobacteria. *Life (Basel, Switzerland)*, 5(1), 164–180.
<https://doi.org/10.3390/life5010164>
- 41. Kieft, K., Adams, A., Salamzade, R., Kalan, L., Anantharaman, K. (2022).
*vRhyme enables binning of viral genomes from metagenomes* | *Nucleic Acids*
*Research* | Oxford Academic.
<https://academic.oup.com/nar/article/50/14/e83/6584432>
- 42. Kieft, K., Zhou, Z., & Anantharaman, K. (2019). VIBRANT: Automated recovery,
annotation and curation of microbial viruses, and evaluation of virome function
from genomic sequences. *bioRxiv*, 855387. <https://doi.org/10.1101/855387>
- 43. Kieft, K., Zhou, Z., Anderson, R. E., Buchan, A., Campbell, B. J., Hallam, S. J.,
Hess, M., Sullivan, M. B., Walsh, D. A., Roux, S., & Anantharaman, K. (2021).
Ecology of inorganic sulfur auxiliary metabolism in widespread bacteriophages.
*Nature Communications*, 12(1), Article 1. [https://doi.org/10.1038/s41467-021-](https://doi.org/10.1038/s41467-021-23698-5)
[23698-5](https://doi.org/10.1038/s41467-021-23698-5)
- 44. Kim, J.-G., Park, S.-J., Sinninghe Damsté, J. S., Schouten, S., Rijpstra, W. I. C.,
Jung, M.-Y., Kim, S.-J., Gwak, J.-H., Hong, H., Si, O.-J., Lee, S., Madsen, E. L.,
& Rhee, S.-K. (2016). Hydrogen peroxide detoxification is a key mechanism for
growth of ammonia-oxidizing archaea. *Proceedings of the National Academy of*

- *Sciences of the United States of America*, 113(28), 7888–7893.
<https://doi.org/10.1073/pnas.1605501113>
- 45. Kim M, Shin B, Lee J, Park HY, Park W. Culture-independent and culture-
dependent analyses of the bacterial community in the phycosphere of
cyanobloom-forming *Microcystis aeruginosa*. *Sci Rep*. 2019 Dec 31;9(1):20416.
<https://pubmed.ncbi.nlm.nih.gov/31892695/>
- 46. Kimura, S., Sako, Y., & Yoshida, T. (2012). Rapid *Microcystis* cyanophage gene
diversification revealed by long- and short-term genetic analyses of the tail
sheath gene in a natural pond. *Applied and Environmental Microbiology*, 79(8),
2789–2795. <https://doi.org/10.1128/AEM.03751-12>
- 47. Kimura S, Uehara M, Morimoto D, Yamanaka M, Sako Y, Yoshida T. Incomplete
Selective Sweeps of *Microcystis* Population Detected by the Leader-End
CRISPR Fragment Analysis in a Natural Pond. *Front Microbiol*. 2018 Mar
8;9:425. <https://pubmed.ncbi.nlm.nih.gov/29568293/>
- 48. Kimura-Sakai S, Sako Y, Yoshida T. Development of a real-time PCR assay for
the quantification of Ma-LMM01-type *Microcystis* cyanophages in a natural pond.
*Lett Appl Microbiol*. 2015 Apr;60(4):400-8.
<https://pubmed.ncbi.nlm.nih.gov/25580646/>
- 49. Koskella, B., & Brockhurst, M. A. (2014). Bacteria-phage coevolution as a driver
of ecological and evolutionary processes in microbial communities. *FEMS*
*Microbiology Reviews*, 38(5), 916–931. <https://doi.org/10.1111/1574-6976.12072>
- 50. Laczny, C. C., Sternal, T., Plugaru, V., Gawron, P., Atashpendar, A., Margossian,
H. H., Coronado, S., der Maaten, L. van, Vlassis, N., & Wilmes, P. (2015).

- VizBin—An application for reference-independent visualization and human-
augmented binning of metagenomic data. *Microbiome*, 3(1), 1.
<https://doi.org/10.1186/s40168-014-0066-1>
- 51. Langmead, B., & Salzberg, S. L. (2012). Fast gapped-read alignment with Bowtie
2. *Nature Methods*, 9(4), Article 4. <https://doi.org/10.1038/nmeth.1923>
- 52. Li, D., Liu, C.-M., Luo, R., Sadakane, K., & Lam, T.-W. (2015). MEGAHIT: An
ultra-fast single-node solution for large and complex metagenomics assembly via
succinct de Bruijn graph. *Bioinformatics*, 31(10), 1674–1676.
<https://doi.org/10.1093/bioinformatics/btv033>
- 53. Li H. and Durbin R. (2009) Fast and accurate short read alignment with Burrows-
Wheeler Transform. *Bioinformatics*, 25:1754-60. [PMID: 19451168]
- 54. Li, H., Handsaker, B., Wysoker, A., Fennell, T., Ruan, J., Homer, N., Marth, G.,
Abecasis, G., Durbin, R., & 1000 Genome Project Data Processing Subgroup.
(2009). The Sequence Alignment/Map format and SAMtools. *Bioinformatics*
(*Oxford, England*), 25(16), 2078–2079.
<https://doi.org/10.1093/bioinformatics/btp352>
- 55. Li, H., Seqtk: a fast and lightweight tool for processing FASTA or FASTQ
sequences, 2013. <http://github.com/lh3/seqtk>
- 56. Liao, Y., Smyth, G. K., & Shi, W. (2013). The Subread aligner: Fast, accurate and
scalable read mapping by seed-and-vote. *Nucleic Acids Research*, 41(10), e108.
<https://doi.org/10.1093/nar/gkt214>
- 57. Liao, Y., Smyth, G. K., & Shi, W. (2014). featureCounts: An efficient general
purpose program for assigning sequence reads to genomic features.

- *Bioinformatics (Oxford, England)*, 30(7), 923–930.
- <https://doi.org/10.1093/bioinformatics/btt656>
- 58. Lin, W., Li, D., Sun, Z., Tong, Y., Yan, X., Wang, C., Zhang, X., & Pei, G. (2020).
- A novel freshwater cyanophage vB_MelS-Me-ZS1 infecting bloom-forming
- cyanobacterium *Microcystis elabens*. *Molecular Biology Reports*, 47(10), 7979–
- 7989. <https://doi.org/10.1007/s11033-020-05876-8>
- 59. Malki, K., Kula, A., Bruder, K., Sible, E., Hatzopoulos, T., Steidel, S., Watkins, S.
- C., & Putonti, C. (2015). Bacteriophages isolated from Lake Michigan
- demonstrate broad host-range across several bacterial phyla. *Virology Journal*,
- 12(1), 164. <https://doi.org/10.1186/s12985-015-0395-0>
- 60. McDaniel, L. D., Young, E., Delaney, J., Ruhnau, F., Ritchie, K. B., & Paul, J. H.
- (2010). High frequency of horizontal gene transfer in the oceans. *Science (New*
- *York, N.Y.)*, 330(6000), 50. <https://doi.org/10.1126/science.1192243>
- 61. McKindles, K. M., Manes, M. A., DeMarco, J. R., McClure, A., McKay, R. M.,
- Davis, T. W., & Bullerjahn, G. S. (2020). Dissolved Microcystin Release
- Coincident with Lysis of a Bloom Dominated by *Microcystis* spp. In Western Lake
- Erie Attributed to a Novel Cyanophage. *Applied and Environmental Microbiology*.
- <https://doi.org/10.1128/AEM.01397-20>
- 62. Michalek et al. (2013). *Record-setting algal bloom in Lake Erie caused by*
- *agricultural and meteorological trends consistent with expected future conditions*
- *| PNAS*. <https://www.pnas.org/doi/full/10.1073/pnas.1216006110>
- 63. Monchamp ME, Pick FR, Beisner BE, Maranger R. Nitrogen forms influence
- microcystin concentration and composition via changes in cyanobacterial

- community structure. PLoS One. 2014 Jan 10;9(1):e85573. doi:
10.1371/journal.pone.0085573. PMID: 24427318; PMCID: PMC3888438.
- 64. Morimoto, D., Yoshida, N., Sasaki, A., Nakagawa, S., Sako, Y., & Yoshida, T.
(2023). Ecological Dynamics of Broad- and Narrow-Host-Range Viruses Infecting
the Bloom-Forming Toxic Cyanobacterium *Microcystis aeruginosa*. *Applied and*
*Environmental Microbiology*, 89(2), e02111-22.
<https://doi.org/10.1128/aem.02111-22>
- 65. Morohoshi, T., Maruo, T., Shirai, Y., Kato, J., Ikeda, T., Takiguchi, N., Ohtake, H.,
& Kuroda, A. (2002). Accumulation of inorganic polyphosphate in phoU mutants
of *Escherichia coli* and *Synechocystis* sp. Strain PCC6803. *Applied and*
*Environmental Microbiology*, 68(8), 4107–4110.
<https://doi.org/10.1128/AEM.68.8.4107-4110.2002>
- 66. Naknaen, A., Suttinun, O., Surachat, K., Khan, E., & Pomwised, R. (2021). A
Novel Jumbo Phage PhiMa05 Inhibits Harmful *Microcystis* sp. *Frontiers in*
*Microbiology*, 12. <https://www.frontiersin.org/articles/10.3389/fmicb.2021.660351>
- 67. Nayfach, S., Camargo, A. P., Schulz, F., Eloie-Fadrosh, E., Roux, S., & Kyrpides,
938 N. C. (2021). CheckV assesses the quality and completeness of metagenome-
939 assembled viral genomes. *Nature Biotechnology*, 39(5), 578–585.
<https://doi.org/10.1038/s41587-020-00774-7>
- 68. NCBI. (2019, April 2). *BLAST+ v2.9.0*. NCBI Insights.
[https://ncbiinsights.ncbi.nlm.nih.gov/2019/04/02/blast-2-9-0-now-available-with-](https://ncbiinsights.ncbi.nlm.nih.gov/2019/04/02/blast-2-9-0-now-available-with-enhanced-support-for-new-database-format-and-improved-performance/)
[enhanced-support-for-new-database-format-and-improved-performance/](https://ncbiinsights.ncbi.nlm.nih.gov/2019/04/02/blast-2-9-0-now-available-with-enhanced-support-for-new-database-format-and-improved-performance/)

- 69. O'neil, J.M., Davis, T.W., Buford, M.A. & Gobler, C.J. (2012). *The rise of harmful*
*cyanobacteria blooms: The potential roles of eutrophication and climate*
*change—ScienceDirect.*
<https://www.sciencedirect.com/science/article/pii/S1568988311001557>
- 70. Ou, T, Li S, Liao X, Zhang Q. Cultivation and characterization of the MaMV-DC
cyanophage that infects bloom-forming cyanobacterium *Microcystis aeruginosa*.
*Virol Sin.* 2013 Oct;28(5):266-71. doi: 10.1007/s12250-013-3340-7. Epub 2013
Aug 26. PMID: 23990146; PMCID: PMC8208409.
- 71. Paerl, H. W. and Huisman, J. (2008). *Blooms Like It Hot | Science.*
<https://www.science.org/doi/10.1126/science.1155398>
- 72. Paerl, H. W. and Huisman, J. (2009). Climate change: A catalyst for global
expansion of harmful cyanobacterial blooms. *Environmental Microbiology*
*Reports*, 1(1), 27–37. <https://doi.org/10.1111/j.1758-2229.2008.00004.x>
- 73. Parks, D. H., Imelfort, M., Skennerton, C. T., Hugenholtz, P., & Tyson, G. W.
(2015). CheckM: Assessing the quality of microbial genomes recovered from
isolates, single cells, and metagenomes. *Genome Research*, 25(7), 1043–1055.
<https://doi.org/10.1101/gr.186072.114>
- 74. Pérez-Carrascal, O. M., Terrat, Y., Giani, A., Fortin, N., Greer, C. W., Tromas, N.,
& Shapiro, B. J. (2019). Coherence of *Microcystis* species revealed through
population genomics. *The ISME Journal*, 13(12), 2887–2900.
<https://doi.org/10.1038/s41396-019-0481-1>

- 75. Poulos, B., John, S., Sullivan, M.B. (2017). Iron Chloride Flocculation of
Bacteriophages from Seawater. *Bacteriophages*, 1681.
https://link.springer.com/protocol/10.1007/978-1-4939-7343-9_4
- 76. Pound, H. L., & Wilhelm, S. W. (2020). Tracing the active genetic diversity of
*Microcystis* and *Microcystis* phage through a temporal survey of Taihu. *PLoS*
*ONE*, 15(12), e0244482. <https://doi.org/10.1371/journal.pone.0244482>
- 77. Preece, E. P., Hardy, F. J., Moore, B. C., & Bryan, M. (2017). A review of
microcystin detections in Estuarine and Marine waters: Environmental
implications and human health risk. *Harmful Algae*, 61, 31–45.
<https://doi.org/10.1016/j.hal.2016.11.006>
- 78. Qian, et al. (2022). *Viruses* | Free Full-Text | A Novel Freshwater Cyanophage,
*Mae-Yong924-1*, Reveals a New Family. [https://www-mdpi-](https://www-mdpi-com.proxy.lib.umich.edu/1999-4915/14/2/283)
[com.proxy.lib.umich.edu/1999-4915/14/2/283](https://www-mdpi-com.proxy.lib.umich.edu/1999-4915/14/2/283)
- 79. Ren, J., Ahlgren, N. A., Lu, Y. Y., Fuhrman, J. A., & Sun, F. (2017). VirFinder: A
novel k-mer based tool for identifying viral sequences from assembled
metagenomic data. *Microbiome*, 5(1), 69. [https://doi.org/10.1186/s40168-017-](https://doi.org/10.1186/s40168-017-0283-5)
[0283-5](https://doi.org/10.1186/s40168-017-0283-5)
- 80. Rieck, A., Herlemenn, D., Jurgens, K., Grossart, H.P. (2015). Particle-Associated
Differ from Free-Living Bacteria in Surface Waters of the Baltic Sea.
[https://www.frontiersin.org/journals/microbiology/articles/10.3389/fmicb.2015.012](https://www.frontiersin.org/journals/microbiology/articles/10.3389/fmicb.2015.01297/full)
[97/full](https://www.frontiersin.org/journals/microbiology/articles/10.3389/fmicb.2015.01297/full)
- 81. Rosenwasser, S., Ziv, C., Creveld, S. G. van, & Vardi, A. (2016). Virocell
Metabolism: Metabolic Innovations During Host-Virus Interactions in the Ocean.

- *Trends in Microbiology*, 24(10), 821–832.
- <https://doi.org/10.1016/j.tim.2016.06.006>
- 82. Roux and Bolduc. (2015). *MAVERICLab / stampede-cluster-genomes—Bitbucket*.
- <https://bitbucket.org/MAVERICLab/stampede-cluster-genomes/src/master/>
- 83. Roux, S., Adriaenssens, E. M., Dutilh, B. E., Koonin, E. V., Kropinski, A. M.,
- Krupovic, M., Kuhn, J. H., Lavigne, R., Brister, J. R., Varsani, A., Amid, C., Aziz,
- R. K., Bordenstein, S. R., Bork, P., Breitbart, M., Cochrane, G. R., Daly, R. A.,
- Desnues, C., Duhaime, M. B., ... Elie-Fadrosh, E. A. (2019). Minimum
- Information about an Uncultivated Virus Genome (MIUViG). *Nature*
- *Biotechnology*, 37(1), Article 1. <https://doi.org/10.1038/nbt.4306>
- 84. Roux, S., Enault, F., Hurwitz, B. L., & Sullivan, M. B. (2016). VirSorter: Mining
- viral signal from microbial genomic data. *PeerJ*, 3, e985.
- <https://doi.org/10.7717/peerj.985>
- 85. Schmidt, M.L., Bopaiah A Biddanda, Anthony D Weinke, Edna Chiang, Fallon
- Januska, Ruben Props, Vincent J Denef, Microhabitats are associated with
- diversity–productivity relationships in freshwater bacterial communities, *FEMS*
- *Microbiology Ecology*, Volume 96, Issue 4, April 2020, fiaa029,
- <https://doi.org/10.1093/femsec/fiaa029>
- 86. Schuurmans JM, Brinkmann BW, Makower AK, Dittmann E, Huisman J, Matthijs
- HCP. Microcystin interferes with defense against high oxidative stress in harmful
- cyanobacteria. *Harmful Algae*. 2018 Sep;78:47-55. doi:
- 10.1016/j.hal.2018.07.008. Epub 2018 Aug 10. PMID: 30196924.

- 87. Schwengers, O, Jelonek, L, Dieckmann, MA, Beyvers, S, Blom, J, Goesmann, A.
Bakta: rapid and standardized annotation of bacterial genomes via alignment-
free sequence identification. *Microb Genom.* 2021 Nov;7(11):000685.
<https://doi.org/10.1099/mgen.0.000685>
- 88. Shaffer et al. (2020). *DRAM for distilling microbial metabolism to automate the*
*curation of microbiome function | Nucleic Acids Research | Oxford Academic.*
<https://academic.oup.com/nar/article/48/16/8883/5884738>
- 89. Smith, D. J., Tan, J. Y., Powers, M. A., Lin, X. N., Davis, T. W., & Dick, G. J.
(2021). Individual *Microcystis* colonies harbour distinct bacterial communities that
differ by *Microcystis* oligotype and with time. *Environmental Microbiology*, 23(6),
3020–3036. <https://doi.org/10.1111/1462-2920.15514>
- 90. Soucy, S. M., Huang, J., & Gogarten, J. P. (2015). Horizontal gene transfer:
Building the web of life. *Nature Reviews Genetics*, 16(8), Article 8.
<https://doi.org/10.1038/nrg3962>
- 91. Steffen, MM, Belisle BS, Watson SB, Boyer GL, Bourbonniere RA, Wilhelm SW.
2015. Metatranscriptomic Evidence for Co-Occurring Top-Down and Bottom-Up
Controls on Toxic Cyanobacterial Communities. *Appl Environ Microbiol* 81:.
<https://doi.org/10.1128/AEM.04101-14>
- 92. Steffen, M. M., Davis, T. W., McKay, R. M., Bullerjahn, G. S., Krausfeldt, L. E.,
Stough, J. M. A., Neitzey, M. L., Gilbert, N. E., Boyer, G. L., Johengen, T. H.,
Gossiaux, D. C., Burtner, A. M., Rowe, M., Dick, G. J., Meyer, K., Levy, S.,
Boone, B., Wynne, T., Zimba, P. V., ... Wilhelm, S. W. (2017). Ecophysiological

- examination of the Lake Erie *Microcystis* bloom in 2014: Linkages between
biology and the water supply shutdown of Toledo, Ohio. *Environmental Science*.
- 93. Suttle, C. A. (2007). Marine viruses—Major players in the global ecosystem.
*Nature Reviews. Microbiology*, 5(10), 801–812.
<https://doi.org/10.1038/nrmicro1750>
- 94. Suzuki K, Yoshida K, Nakanishi Y, Fukuda S. An equation-free method reveals
the ecological interaction networks within complex microbial ecosystems.
*Methods Ecol Evol*. 2017; 8: 1774–1785. [https://doi.org/10.1111/2041-](https://doi.org/10.1111/2041-210X.12814)
[210X.12814](https://doi.org/10.1111/2041-210X.12814)
- 95. Takashima, Y., Yoshida, T., Yoshida, M., Shirai, Y., Tomaru, Y., Takao, Y.,
Hiroishi, S., & Nagasaki, K. (2007). Development and Application of Quantitative
Detection of Cyanophages Phylogenetically Related to Cyanophage Ma-LMM01
Infecting *Microcystis aeruginosa* in Fresh Water. *Microbes and Environments*,
22(3), 207–213. <https://doi.org/10.1264/jsme2.22.207>
- 96. Tonkin-Hill, G., MacAlasdair, N., Ruis, C. et al. Producing polished prokaryotic
pangenomes with the Panaroo pipeline. *Genome Biol* 21, 180 (2020).
<https://doi.org/10.1186/s13059-020-02090-4>
- 97. Tromas, N., Taranu, ZE, Martin, BD, Willis, A, Fortin, N, Greer, CW, Shapiro, BJ.
Niche Separation Increases With Genetic Distance Among Bloom-Forming
Cyanobacteria. *Front Microbiol*. 2018 Mar 27;9:438.
<https://doi.org/10.3389/fmicb.2018.00438>
- 98. Tucker, S., & Pollard, P. (2005). Identification of Cyanophage Ma-LBP and
Infection of the Cyanobacterium *Microcystis aeruginosa* from an Australian

- Subtropical Lake by the Virus. *Applied and Environmental Microbiology*, 71(2),
629–635. <https://doi.org/10.1128/AEM.71.2.629-635.2005>
- 99. Visser, P. M., Verspagen, J. M. H., Sandrini, G., Stal, L. J., Matthijs, H. C. P.,
Davis, T. W., Paerl, H. W., & Huisman, J. (2016). How rising CO₂ and global
warming may stimulate harmful cyanobacterial blooms. *Harmful Algae*, 54, 145–
159. <https://doi.org/10.1016/j.hal.2015.12.006>
- 100. Wang J, Bai P, Li Q, Lin Y, Huo D, Ke F, Zhang Q, Li T, Zhao J. Interaction
between cyanophage MaMV-DC and eight *Microcystis* strains, revealed by genetic
defense systems. *Harmful Algae*. 2019 May;85:101699. doi:
10.1016/j.hal.2019.101699. Epub 2019 Nov 8. PMID: 31810530.
- 101. Wang, F., Li, D., Cai, R., Pan, L., Zhou, Q., Liu, W., Qian, M., & Tong, Y. (2022).
A Novel Freshwater Cyanophage Mae-Yong1326-1 Infecting Bloom-Forming
Cyanobacterium *Microcystis aeruginosa*. *Viruses*, 14(9), 2051.
<https://doi.org/10.3390/v14092051>
- 102. Wanner, B. L. (1993). Gene regulation by phosphate in enteric bacteria. *Journal*
*of Cellular Biochemistry*, 51(1), 47–54. <https://doi.org/10.1002/jcb.240510110>
- 103. Weitz, J. S., & Wilhelm, S. W. (2012). Ocean viruses and their effects on
microbial communities and biogeochemical cycles. *F1000 Biology Reports*, 4, 17.
<https://doi.org/10.3410/B4-17>
- 104. Wilson, A. E., Wilson, W. A., & Hay, M. E. (2006). Intraspecific variation in growth
and morphology of the bloom-forming cyanobacterium *Microcystis aeruginosa*.
*Applied and Environmental Microbiology*, 72(11), 7386–7389.
<https://doi.org/10.1128/AEM.00834-06>

- 105. Winter, S. E., Thiennimitr, P., Winter, M. G., Butler, B. P., Huseby, D. L.,
Crawford, R. W., Russell, J. M., Bevins, C. L., Adams, L. G., Tsohis, R. M., Roth, J.
R., & Bäumlér, A. J. (2010). Gut inflammation provides a respiratory electron
acceptor for Salmonella. *Nature*, 467(7314), Article 7314.
<https://doi.org/10.1038/nature09415>
- 106. Wishart, D.S., Han, S., Sukanta Saha, Eponine Oler, Harrison Peters, Jason R
Grant, Paul Stothard, Vasuk Gautam, PHASTEST: faster than PHASTER, better
than PHAST, *Nucleic Acids Research*, Volume 51, Issue W1, 5 July 2023, Pages
W443–W450, <https://doi.org/10.1093/nar/gkad382>
- 107. Xiao, M., Willis, A., & Burford, M. A. (2017). Differences in cyanobacterial strain
responses to light and temperature reflect species plasticity. *Harmful Algae*, 62,
84–93. <https://doi.org/10.1016/j.hal.2016.12.008>
- 108. Yamaguchi H, Suzuki S, Tanabe Y, Osana Y, Shimura Y, Ishida K, Kawachi M.
Complete Genome Sequence of *Microcystis aeruginosa* NIES-2549, a Bloom-
Forming Cyanobacterium from Lake Kasumigaura, Japan. *Genome Announc.* 2015
May 28;3(3):e00551-15. doi: 10.1128/genomeA.00551-15. PMID: 26021928;
PMCID: PMC4447913.
- 109. Yancey, C. E., Smith, D. J., Den Uyl, P. A., Mohamed, O. G., Yu, F., Ruberg, S.
1096 A., Chaffin, J. D., Goodwin, K. D., Tripathi, A., Sherman, D. H., & Dick, G. J.
(2022). Metagenomic and Metatranscriptomic Insights into Population Diversity of
*Microcystis* Blooms: Spatial and Temporal Dynamics of mcy Genotypes, Including
a Partial Operon That Can Be Abundant and Expressed. *Applied and*

[revised manuscript text omitted]

Response to Reviewers 7-17-25

Reviewer Comments = **This font**

Original manuscript lines = This font

Author Response = **This font**

Reviewer #1 Minor Comments

1. Gene clustering analysis (Fig 1G, lines 144-147, 317-321: I think the clustering analysis needs a bit of clarification. How were "core" genes defined? If their presence/absence is meaningful (as implied by Fig 1G), then are they actually core genes? My understanding is that core genes are universally shared among the group (though not always universally detected, due to incomplete MAGs). The methods describe a 90% clustering cutoff, in which case some core protein families could conceivably be split into different sequence clusters, and maybe that's the presence/absence that's being plotted in 1G? In that case, it really is more a reflection of sequence similarity (i.e. another proxy for ANI) than for gene content.

We appreciate the reviewer's insights regarding our *Microcystis* gene clustering analysis and Figure 1G. In response, we (i) more clearly and conventionally defined the gene sets used, and (ii) evaluated the likelihood of the gene clusters having been split into clusters of homologues. We updated the Main Text to reflect these changes.

First, rather than referring to the gene clusters in Figure 1G as 'core gene clusters', we now refer to them as 'pangenome gene clusters', where the pangenome represents the complete set of genes found across all genomes within a given population. Here we considered the pangenome as subdivided into the core genome (genes present in all genomes), the accessory genome (genes present in some, but not all, genomes), and unique genes (those found in only a single genome). Given this framework, updated Figure 1G displays the distribution of the accessory pangenome gene clusters. We have updated this phrasing in the revised manuscript.

Second, as the reviewer pointed out, Panaroo's 90% clustering threshold is sequence identity-based and can lead to highly similar gene variants forming separate

homologous sub-clusters of the same gene. At the reviewer's caution, we sought to determine the extent to which this occurred in our dataset, versus the frequency of bonafide fraction-specific genes that do not arise from split clusters of homologues.

Panaroo's internal quality metrics gave us confidence that splitting of homologues genes into separate clusters was a rare event: the 'Average number of sequences per isolate' metric was consistently low (<5% of clusters having values >1, with an average of 1.14 for those cases), indicating limited overclustering. Furthermore, the 'Non-unique gene name' metric, a direct indicator of potential gene-splitting or mis-annotation, showed only 330 duplicates (~3.8%) out of 8,600 total gene clusters. These metrics gave us confidence that pervasive gene splitting or overclustering, which could confound gene content interpretation, was not a widespread issue in our study.

In this follow-up analysis, we found that the PCA conducted on all *Microcystis* pangenome gene clusters captured considerable variance in its first two principal components (PC1+PC2 explained 44.7% total variance; new Fig. S2). In the new analysis of the *Microcystis* accessory and unique gene clusters only (no core genes), the first PCs yielded even higher explanatory power (PC1+PC2 explained 46.8% total variance; new Fig. 1G). This suggests that the variation explained by distributions of accessory genes (7,593 out of 8,600 pangenome gene clusters, representing 88.3%) and unique genes (266 out of 8,600 pangenome gene clusters, representing 3.1%) have a large impact on the observed gene-based ordination. Our findings align with the prior reports that the accessory and unique genes constitute the vast majority (88-93%, Cai et al., 2023) of the *Microcystis* pangenome. Therefore, the distinct grouping along principal component 1 observed in Fig. 1G effectively reflects differences in the gene family clusters that distinguish *Microcystis* populations between size fractions—and could in turn contribute to the fraction-specific viral population structure later discussed.

Relevant updated lines:

“When presence/absence of *Microcystis* MAG pangenome accessory gene clusters (defined as gene families present in at least 90% of the MAGs, Li. et al., 2022) ~~core genes~~ was considered, 33.331.4% of the variation ~~in core gene clusters~~ could be explained by fraction, with the 100-µm MAGs forming a distinct grouping along principle component 1 (Fig. 1G).”

Relevant updated lines:

In the ~~size-fraction~~ principal components analysis of variation among gene clusters by size-fraction, the first two components of the *Microcystis* MAG accessory ~~core~~ gene clusters offered ~~greater~~ similar explanatory power (46.8%; Fig. 1G) as when compared to the full suite of *Microcystis* pangenome gene clusters (44.7%; SI Fig. 2). This

suggested that the gene representation in the flexible portion of the *Microcystis* pangenome corresponds with size fractions, though little of this difference is functionally annotated (SI Table 3). ~~than the full complement of genes, implying that the relatively stable, shared portion of the *Microcystis* genome correlates more closely with observed genetic differentiation.~~ This finding is consistent with a classification framework recently proposed by Cai et al., (2023), ~~in which phylogenies derived from core genes represent evolutionary trajectories and speciation processes in which the variable accessory genome reflects the ecological adaptations of *Microcystis*,~~ potentially ~~shaping~~ ~~underlying~~ how *Microcystis* lineages interact with both biological and physicochemical components of their environments.

2. I still wonder what genes distinguish the MAGs from different size fractions... though I recognize that maybe the authors don't want to delve into that here because it's a bigger question. But maybe a sentence or two would be nice.

We agree with the reviewer that *Microcystis* genes unique to the different size fractions is intriguing, we also agree that it is another story in itself. We (i) evaluated and summarized the fraction-specific gene content and, per the suggestion, (ii) added text that lists a few of the unique *Microcystis* genes found in two size fractions, and (iii) included a new SI Table (SI Table 3) of these results.

When presence/absence of *Microcystis* MAG pangenome accessory gene clusters (defined as gene families present in at least 90% of the MAGs, Li. et al., 2022) was considered, 33.3% of the variation could be explained by fraction, with the 100- μ m MAGs forming a distinct grouping along principle component 1 (Fig. 1G).

In terms of the distribution of the pangenome gene clusters, a large proportion, 6,738 (78.3%), were shared in at least two fractions, and 3,529 (41%) were shared across all fractions. Some gene clusters were found in only one fraction: 1,572 (18.3% of total clusters) in the 100 μ m fraction only, 12 (0.14%) in 53 μ m, 133 (1.5%) in 3 μ m, and 145 (1.7%) in >0.22 μ m. Most of these fraction-specific genes are not functionally annotated; however, the two genes with functional annotations present in all 100- μ m fractions were *lprI* (lysozyme inhibition) and *menH* (menaquinone biosynthesis; SI Table 3). The only gene with functional annotation in the not-colony-associated-fraction was a restriction enzyme, *awlI* (SI Table 3).

3. Network connections (lines 178-182): just to clarify -- are the 793 vOTUs here limited to only those that are associated with *Microcystis* (and sometimes another

bacterium as well)? Since other bacteria are shown, I was wondering about vOTUs linked to these other bacteria -- maybe just a sentence stating that there were other vOTUs discovered that did not associate with *Microcystis* (if this is true), but they are not discussed here?

The reviewer's assumption is correct in that we focused on only those vOTUs predicted to infect AT LEAST *Microcystis* MAGs were included. We included all other hosts these vOTUs were also predicted to infect. We have clarified this in the updated text.

Relevant updated lines:

We found that at the August 4 toxic bloom peak, 2,026 virus-host pairs were predicted between 454 vOTUs and 17 bacterial MAGs (9 of which were *Microcystis* MAGs; SI Table 1) (Fig. 3A). On the September 29 non-toxic bloom peak, 1,995 virus-host pairs were predicted between 339 vOTUs and 24 bacterial MAGs (8 *Microcystis* MAGs; SI Table 1; Fig. 3B; SI Fig. 4). 97% of viruses were predicted to infect at least one host (SI Fig. 5). A total of 793 putative *Microcystis* vOTUs were identified at the bloom peaks (Fig. 3A-B). These 793 vOTUs represent those predicted to infect at least one *Microcystis* MAG; if these were also predicted to infect any non-*Microcystis* bacterial hosts, these hosts were also included in the network. vOTUs not predicted to infect *Microcystis* were not included in this study.

4. line 224-5 incomplete sentence

Relevant updated lines:

~~While on~~ On September 29, the most abundant *Myoviridae* myovirus vOTUs belonged to the >0.22- μ m not-colony-associated fraction.

5. line 239-240: at different sites? "peak" implies singularity, so it took me a minute to understand how they could peak at two times.

We recognize the reviewer's point on the potential confusion caused by using the word "peak" which is often considered to describe a single event. As a result, we have modified the text to better describe our findings.

Relevant updated lines:

"The highest relative abundance of abundant vOTUs ~~generally peaked~~ were typically observed in either the colony-associated or unassociated fractions during the August 4th and September 29th bloom events ~~on Aug 4 or Sept 29~~ (Fig. 4D), with only three of

the 10 vOTUs ~~peaking at both dates~~ found to be highly abundant across both dates (Fig. 4D).”

6. Fig 6B: maybe color the points by virus family instead of date, since date is shown on the axes? This would make it clear whether the 3 families are more/less abundant on the two dates.

We agree with the reviewer. Point color now reflects the assigned viral family (myovirus, siphovirus, podovirus), indicating their relative abundance on each date. Shape continues to show which of the bloom peaks they were found in, including those found at both dates (new Fig. 6B).

Reviewer #2 Minor Comments

1. The name of the host is not italicized when used to describe the virus (*Microcystis* virus), so this should be edited throughout.

We appreciate the reviewer's correction. We have consulted the ICTV FAQs for guidance on naming convention (<https://ictv.global/faqs>) and have updated the text throughout to correspond with current recommendations.

Per the FAQ we read “A virus name should never be italicized, even when it includes the name of a host species or genus, and should be written in lower case.” As such, when referring to specific vOTUs, they are now referred to as, e.g., Microcystis vOTU 4—no italics. When a bonafide species name is used as a reference, such as *Fukuivirus LMM01* (species name of Microcystis virus Ma-LMM01), the whole name is italicized. When the virus name is used (not species name, as in example in prior sentence), no italics are used.

However, nowhere in the FAQ were we able to find guidance as to when the host reference is used as an adjective to describe the collection of viruses infecting *Microcystis*, rather than the name of a virus or specific vOTU; this is not the same as when a host genus is used to distinguish the name of a single virus. There is reference to “A virus name should never be italicized, even when it includes the name of a host species or genus, and should be written in lower case.” This is not the same as when we are using host taxa name *Microcystis* as an adjective to reference the whole collection of viruses infecting the *Microcystis* host of interest, e.g., *Microcystis* viruses, *Microcystis* viral diversity or even *Microcystis* vOTUs, when we are referring to the

collection of vOTUs predicted to infect our host taxon of interest. Here we continue to italicize *Microcystis*.

In all instances, we want to get this right and have tried to interpret and apply ICTV convention as best as possible. We welcome further guidance.

2. ICTV revised the taxonomical approach to bacteriophage in 2022. The taxonomy of phage has moved to a genome-based approach, and names based on morphology are no longer accepted, meaning, the names Myoviridae, Siphoviridae, and Podoviridae are no longer valid taxonomic groups. The organisms in these groups are now all reassigned to the class Caudoviricetes. In this MS, the taxonomic assignment of viral populations was estimated using Phage Taxonomy Tool, and in this case specific gene markers were chosen to differentiate between these old morphological groups of phages. While there are clearly differences in abundance, host-virus pairs, etc., between the different morphological groups, the current grouping into defunct groups is problematic. I'm not sure what is the best approach, but suggest the analysis could be left as is, and the names of the groups be revised to myovirus, podovirus and siphovirus to indicate they are morphological and not taxonomical groups.

We acknowledge the reviewer's concern with the use of these defunct taxonomic groups; the taxonomic analysis was admittedly performed before this reclassification was easy to integrate into our analyses. In this portion of our study, we were struck by the dynamics of these annotated gene functions, rather than taxonomy outright. We heed the reviewer's guidance and revise the nomenclature, such that 'myovirus,' 'siphovirus,' and 'podovirus' (lowercase, non-italicized) is used throughout the manuscript to indicate their morphological rather than taxonomic classification.

For the methods section, we've included the following text to clarify our approach:

“While the ICTV taxonomy has shifted to a genome-based approach, for the purposes of this study, we refer to 'myovirus', 'siphovirus', and 'podovirus' to denote morphological groupings based on diagnostic genes.”

3. Around line 164 it is suggested that tail sheaths are indicative of lytic phages, but temperate phages can switch between lytic and lysogenic strategies, so they also have tail sheath proteins. Likewise, it is suggested that only temperate phages will have lysis inhibition proteins, but lytic phages can also have lysis inhibition proteins in order to delay/regulate host cell lysis. These genes are therefore not good indicators of viral replication strategy.

We agree with the reviewer and appreciate the correction. We have revised the interpretation in the manuscript to avoid definitive statements about replication strategy based solely on these gene markers.

Relevant updated lines:

“While Ma-LEF01 ~~has genes characteristic of~~ carries genes that have been previously associated with both lytic (e.g., viral tail sheath) and lysogenic (e.g., putative phage anti-repressors, site-specific recombinase, resolvase, lysis inhibition proteins rIIA and B) replication strategies, ~~these genes are not always definitive indicators of a specific viral replication strategy.~~ Nevertheless, neither Ma-LEF01 nor its relatives were identified as integrated prophages in the 50 bacterial MAGs reconstructed in this study.”

4. The section around line 470 attempts to suggest viral replication strategy based on the old morphological groupings of phage. I would suggest this is inappropriate and should be avoided.

We agree with the reviewer's caution. We have revised the discussion section to remove or rephrase any statements that directly link the (now referred to as) 'myovirus', 'siphovirus', and 'podovirus' categories to definitive lytic or lysogenic replication strategies.

Relevant updated lines:

Temporal and spatial patterns of viral marker genes offer clues about how *Microcystis* viruses influence and adapt to cHAB phases. Pound et al. (2020) observed ~~viral marker genes in a 2014 *Microcystis* bloom in Lake Tai (China) that Myoviruses were associated with early season host genotypes, while Siphoviruses became more prominent later in the bloom~~ in a 2014 *Microcystis* bloom in Lake Tai (China) marker genes they associated with myoviruses were more prevalent among early-season host genotypes, while those associated with siphoviruses were more prominent later in the bloom. Similarly, our study found the distribution of ~~Myoviridae (T4-like), Siphoviridae (lambda-like), and Podoviridae (T7-like) marker genes shifted across bloom phases~~ marker genes associated with myovirus (T4-like), siphovirus (lambda-like), and podovirus (T7-like) shifted across bloom phases. While Pound et al. reported a lytic-to-lysogenic shift in marker gene abundances, ~~we found a temporal transition between lytic types (e.g., T4-like Myoviruses dominant on August 4 and T7-like Podoviruses more abundant by September 29) and a stable presence of abundant Siphoviruses at each bloom peak.~~ our data showed a temporal transition in the prevalence of diagnostic genes associated with T4-like myoviruses (dominant on

August 4) and T7-like podoviruses (more abundant by September 29), alongside a stable presence of diagnostic genes associated with siphoviruses at each bloom peak. However, viral ~~taxonomic-family-morphological~~ designation can offer only general clues about whether a virus is more likely to be virulent or temperate; it is not a definitive predictor, especially when relying on sequence data alone.

5. Line 637-639 indicates that integrase was included in the PTT taxonomic analysis as an indicator of lysogenic integration. While most temperate phage analyzed to date have integrases, some do not, so conclusions based on this analysis should be approached with caution.

We acknowledge the reviewer's important clarification regarding integrase genes. We include a new statement in the Methods section to highlight this nuance and encourage cautious interpretation of replication strategies based on integrase alone.

Relevant updated lines:

"We also included integrase, a gene indicative of lysogenic integration, to examine the composition and replication strategies of these viruses at the bloom peaks. It is important to note, however, that while integrases are commonly found in temperate phages, their presence is not universally indicative of lysogenic integration, nor does their absence preclude a temperate lifestyle. Therefore, conclusions regarding replication strategies based solely on integrase presence should be interpreted with caution."

Re: mSystems00575-25R1 (Tracking putative *Microcystis* viruses and virus-host associations across distinct phases of a *Microcystis*-dominated bloom)

Dear Dr. Melissa B. Duhaime:

Thank you for your detailed and thoughtful responses to the reviews. Your manuscript has been accepted, and I am forwarding it to the ASM production staff for publication. Your paper will first be checked to make sure all elements meet the technical requirements. ASM staff will contact you if anything needs to be revised before copyediting and production can begin. Otherwise, you will be notified when your proofs are ready to be viewed.

Cover Image Submissions: If you would like to submit a potential Cover Image, please email a file and a short legend to mSystems@asmusa.org. Please note that we can only consider images that (i) the authors created or own and (ii) have not been previously published. By submitting, you agree that the image can be used under the same terms as the published article. Image File requirements: TIF/EPS, 7.5 inches wide by 8.25 inches tall (at least 2,250 pixels wide by 2,475 pixels tall), minimum 300 dpi resolution (600 dpi preferred), RGB, and no figure elements, e.g., arrows or panel labels. The legend should be a short description of the image, 1-2 sentences recommended. Please download and use this interactive template in Adobe to ensure that your proposed cover image meets our size requirements (<https://journals.asm.org/pb-assets/pdf-text-excel-files/ASM-Interactive-Sizing-Cover-Template-1715689791.pdf>).

Sincerely,
Jeffrey Blanchard
Editor
mSystems